# A high-resolution temporal atlas of the SARS-CoV-2 translatome and transcriptome

Doyeon Kim[1,6], Sukjun Kim [1,6], Joori Park [2,3,6], Hee Ryung Chang[1,6], Jeeyoon Chang [2,3,6], Junhak Ahn [1,6], Heedo Park [4,6], Junehee Park[1], Narae Son[1], Gihyeon Kang[1], Jeonghun Kim [4], Kisoon Kim[4], Man-Seong Park[4✉], Yoon Ki Kim [2,3✉] & Daehyun Baek[1,5✉]

COVID-19 is caused by severe acute respiratory syndrome coronavirus 2 (SARS-CoV-2), which infected >200 million people resulting in >4 million deaths. However, temporal landscape of the SARS-CoV-2 translatome and its impact on the human genome remain unexplored. Here, we report a high-resolution atlas of the translatome and transcriptome of SARS-CoV-2 for various time points after infecting human cells. Intriguingly, substantial amount of SARS-CoV-2 translation initiates at a novel translation initiation site (TIS) located in the leader sequence, termed TIS-L. Since TIS-L is included in all the genomic and sub-genomic RNAs, the SARS-CoV-2 translatome may be regulated by a sophisticated interplay between TIS-L and downstream TISs. TIS-L functions as a strong translation enhancer for ORF S, and as translation suppressors for most of the other ORFs. Our global temporal atlas provides compelling insight into unique regulation of the SARS-CoV-2 translatome and helps comprehensively evaluate its impact on the human genome.

[1] School of Biological Sciences, Seoul National University, Seoul, Republic of Korea. [2] Creative Research Initiatives Center for Molecular Biology of Translation, Korea University, Seoul, Republic of Korea. [3] Division of Life Sciences, Korea University, Seoul, Republic of Korea. [4] Department of Microbiology, Institute for Viral Diseases, College of Medicine, Korea University, Seoul, Republic of Korea. [5] Bioinformatics Institute, Seoul National University, Seoul, Republic of Korea. [6]These authors contributed equally: Doyeon Kim, Sukjun Kim, Joori Park, Hee Ryung Chang, Jeeyoon Chang, Junhak Ahn, Heedo Park. ✉email: manseong.park@gmail.com; yk-kim@korea.ac.kr; baek@snu.ac.kr

Severe acute respiratory syndrome coronavirus 2 (SARS-CoV-2) is a betacoronavirus that belongs to family Coronaviridae along with other human coronavirus species (229E, OC43, NL63, HKU1, SARS-CoV, and MERS-CoV) known to be pathogenic to humans[1]. While the human coronaviruses (HCoV) 229E, OC43, NL63, and HKU1 induce relatively mild respiratory diseases, HCoV SARS-CoV, MERS-CoV, and SARS-CoV-2 have resulted in severe outbreaks in 2002, 2012, and 2019, respectively, with fatal disease outcomes[1,2]. Especially, SARS-CoV-2 has caused an unprecedented scale of a pandemic named the coronavirus disease 2019 (COVID-19)[1,3–5]. As the United Nations has recently declared, COVID-19 is not only a pandemic but also a substantial crisis deeply affecting the societies and economics on a global scale[6,7].

SARS-CoV-2 has a ~30 kb single-stranded positive-sense RNA genome that is used as a template for subgenomic RNA (sgRNA) transcription. Two-thirds of the genome at its 5′ end consist of ORFs 1a and 1b that encode polyprotein 1a (pp1a) and pp1ab, respectively, that are posttranslationally processed to 16 nonstructural proteins[8–10]. The rest of the genome encodes structural and accessory proteins, whose composition varies among the coronaviruses[9]. Recent studies have reported that the overall genome structure of SARS-CoV-2 is similar to rest of the coronaviruses, containing ORFs 1a and 1b and other genes for structural proteins, spike (S), envelope (E), membrane (M), nucleocapsid (N), and accessory proteins, 3a, 6, 7a, 7b, 8, and 10[2,11].

While ORFs 1a and 1b are translated directly from the genomic RNA (gRNA), the rest of the ORFs are translated from sgRNAs. The sgRNAs are nested coterminal sets of positive stranded RNAs that are all 5′ capped and 3′ polyadenylated. The identical 5′ ends are the leader sequence that contains transcription regulatory sequence-leader (TRS-L) at its 3′ ends. The TRSs are also located upstream of each ORF (TRS-body (TRS-B)) and the 6-nt core sequence in the middle is identical for all TRSs. The 5′ end of each ORF is fused with the leader sequence through discontinuous transcription by the RNA-dependent RNA polymerase complex encoded by ORFs 1a and 1b[2,9,12].

Whereas the viral genomic replication and transcription are driven by viral proteins, the translation process relies on the host machinery. Therefore, observation of both viral and human translatome dynamics upon infection is crucial to understanding the molecular mechanism of the SARS-CoV-2 pathogenicity in humans. Although the SARS-CoV-2 transcriptome[11] and translatome[13] have been recently reported, no study has concomitantly provided the temporal landscape of both the viral and human transcriptome and translatome upon infection so far.

In this study, we report a high-resolution temporal atlas of the SARS-CoV-2 translatome and transcriptome for early (0, 1, 2, and 4) in comparison to late (12, 16, 24, and 36) hours post infection (hpi) in a human lung cell line (Calu-3) as well as another temporal atlas of the human translatome and transcriptome in response to SARS-CoV-2 infection. Ribosome-protected mRNA fragment sequencing (RPF-seq) and quantitative profiling of initiating ribosomes sequencing (QTI-seq) were performed to accurately quantitate translation initiation (TI) sites (TISs)[14,15] along with mRNA sequencing (mRNA-seq) and small RNA sequencing (sRNA-seq). Our high-resolution temporal atlas of the translatome and transcriptome of the SARS-CoV-2 and its human host will provide an unprecedented opportunity to comprehensively investigate the SARS-CoV-2 translatome and transcriptome.

## Results

**Generation of massive-scale datasets of the SARS-CoV-2 translatome and transcriptome**. To systematically investigate temporal landscape of the SARS-CoV-2 translatome and transcriptome,

Calu-3 cells were infected with SARS-CoV-2 at multiplicity of infection (MOI) 10. The cells were harvested at 0, 1, 2, and 4 hpi to observe early response and 12, 16, 24, and 36 hpi to observe late response to the viral infection (Fig. 1a). At each time point, we performed RPF-seq, QTI-seq, mRNA-seq, and sRNA-seq. RPF-seq is a ribosome profiling technique where cells are treated with cycloheximide (CHX) to stall elongating ribosomes that occupy on mRNAs[14] (Fig. 1a). After lysis, the lysate is treated with RNase I, and the 28–29-nt mRNA fragments that are protected by ribosomes are sequenced to quantitate the ribosome occupancy on each mRNA to infer its translation rate. QTI-seq is another ribosome profiling technique to more specifically quantitate TI[15], which utilizes harringtonine (Harr) or lactimidomycin instead of CHX to stall initiating ribosomes (Fig. 1a). To compare with other cell lines and MOI values, RPF-seq and QTI-seq were additionally performed with Calu-3 (a human lung cell line), Caco-2 (a human intestine cell line), and Vero (a kidney epithelial cell line of an African green monkey). These three cell lines were infected at MOI 0.1 and were incubated for 48 h (Calu-3 and Caco-2) and for 24 h (Vero).

After removing the reads that were of low quality or unmapped (see Supplementary Fig. 1a, Supplementary Table 1, and "Methods"), we measured the correlation between replicates for RPF-seq, QTI-seq, and mRNA-seq. The correlation was very strong between replicates (Spearman's correlation coefficient, $\rho^2 > 0.80$ for all replicates), which indicates our datasets are highly reliable (Fig. 1b and Supplementary Fig. 1b, c). Utilizing these datasets, the relative fraction of mapped reads onto either the SARS-CoV-2 or human genomes was computed (Fig. 1c and Supplementary Fig. 1a). The gradual increase of viral reads proportion compared to human genome reads over time was detected for RPF-seq, QTI-seq, and mRNA-seq, reaching up to 55% for mRNA-seq at 36 hpi (Fig. 1c). Consistently, the replication kinetics of SARS-CoV-2 analyzed by plaque assay at each time point indicates gradual increase in the number of the viral particles, reaching the maximal titer at 36 hpi in Calu-3 at a high MOI of 10 (Fig. 1d). On the other hand, at both MOIs (10 and 0.1), the virus titer peaked at 24 hpi in Vero. The virus grew at a much faster rate in Vero than in Calu-3 cell lines, which may be attributable to Vero's incapability of synthesizing interferons (IFNs)[16,17].

To assess the quality of our translatome dataset, we examined the length distribution of our RPF-seq and QTI-seq data. From all conditions, we observed a discrete peak at 28–29 nts (Supplementary Fig. 1d), which is the expected size of RPFs[18]. For all of the translatome data, >50% of the reads were mapped to the first position of triplet codon (Supplementary Fig. 1e), and this clear periodicity confirms the enrichment of RPFs[19]. Moreover, we quantitatively assessed the relative distribution of the reads within ORFs (Fig. 1e), which provides the global overview of the viral and human translatome. While the mRNA-seq reads were relatively evenly distributed, RPF-seq and QTI-seq reads clearly exhibited the triplet periodicity aforementioned and were enriched near the TISs of ORFs. The QTI-seq reads were more enriched in the annotated TISs for ORFs compared to RPF-seq reads as expected. Taken together, these results demonstrate our translatome dataset is overall of high quality.

**A high-resolution temporal atlas of the SARS-CoV-2 translatome and transcriptome**. Employing our RPF-seq, QTI-seq, mRNA-seq, and sRNA-seq datasets obtained at various time points post viral infection, a high-resolution temporal atlas of SARS-CoV-2 translatome and transcriptome was constructed (Fig. 2a and Supplementary Fig. 2a–d). At 4 hpi, structural and accessory genes began to be expressed and at 36 hpi, the viral

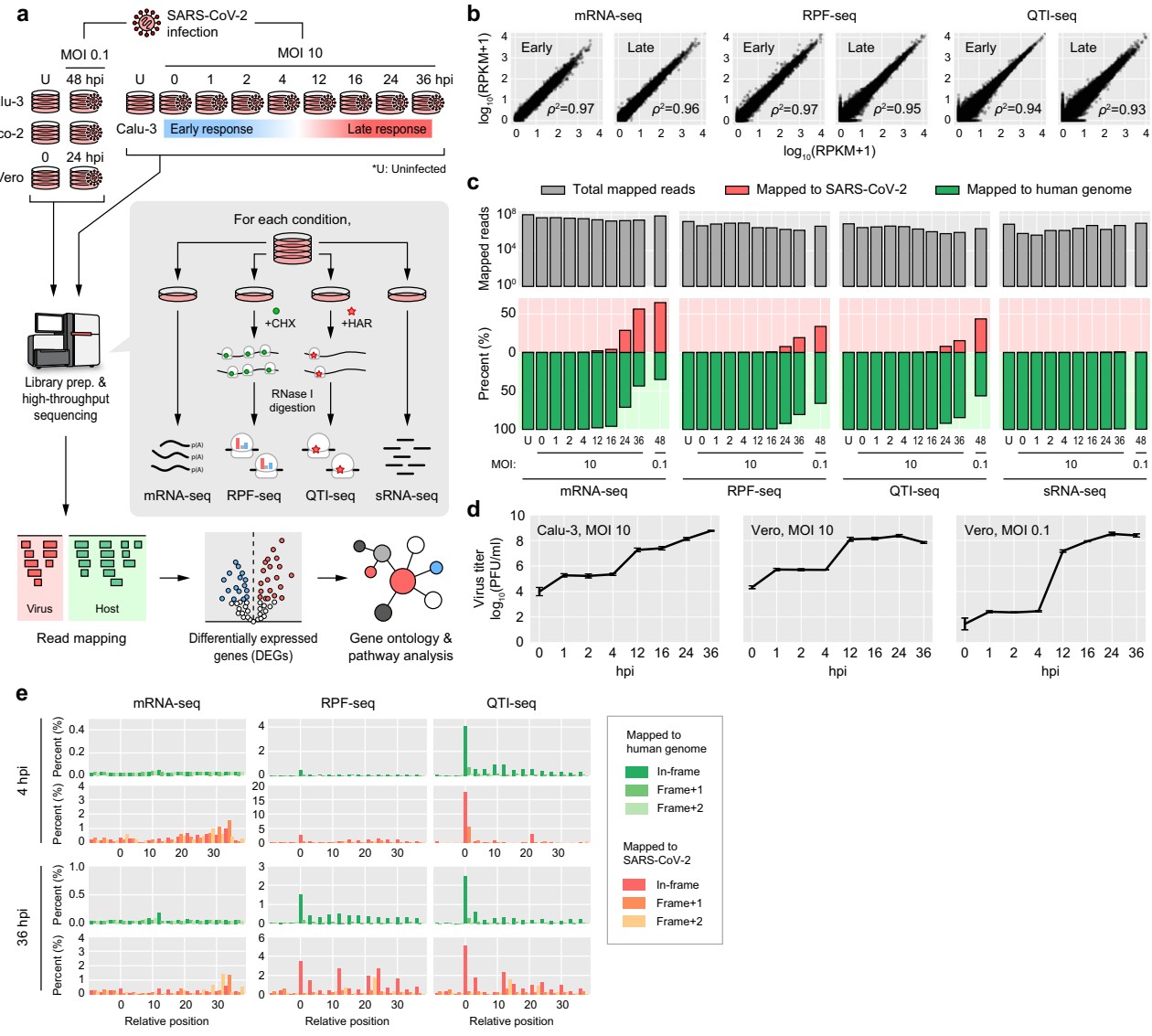

**Fig. 1 Experimental design and generation of massive-scale datasets. a** Overall design of the study. A fine-scale temporal atlas of the SARS-CoV-2 translatome and transcriptome was constructed for early (0, 1, 2, and 4) and late (12, 16, 24, and 36) hours post infection (hpi) at multiplicity of infection (MOI) of 10 and at 48 hpi (MOI = 0.1) in Calu-3 and Caco-2 cell lines and at 24 hpi (MOI = 0.1) in Vero cell line. The temporal atlas of the human translatome and transcriptome in response to SARS-CoV-2 invasion were also constructed. Ribosome-protected mRNA fragment sequencing (RPF-seq), quantitative profiling of initiating ribosomes sequencing (QTI-seq), mRNA sequencing (mRNA-seq), and small RNA sequencing (sRNA-seq) were performed for each time point. For more details, see "Methods". **b** High reproducibility between the replicates of sequenced data. Correlation coefficient (Spearman's $\rho$) was calculated by comparing host gene expression levels between replicates. Both x- and y-axes represent $\log_{10}(RPKM + 1)$. Representative examples from early and late hpi are displayed. For a full version, see Supplementary Fig. 1b, c. **c** The number of total mapped reads are shown in $\log_{10}$ scale (top) with the relative fraction of reads mapped to the human and SARS-CoV-2 genomes (bottom). The upward and downward directions of the y-axis indicate the fraction of the reads mapped to the SARS-CoV-2 and human genomes, respectively. **d** A growth dynamics curve of SARS-CoV-2 in Calu-3 and Vero cell lines after infection. The mean values ±95% confidence intervals are displayed ($n = 3$ biological independent experiments). **e** Distribution of mRNA-seq (left), RPF-seq (middle), and QTI-seq (right) reads with respect to the relative position near the start of ORF. For 4 and 36 hpi, the 13th nucleotide position (12-nt offset from the 5' end) of the reads mapped to human (green) or SARS-CoV-2 (red) was counted for each sequencing data. The relative fraction to the amount of reads mapped to the entire ORF was calculated for each position, and the y-axis represents the average of the relative fractions for ORFs with >50 reads mapped. Open reading frames are depicted as three different colored bars with the darkest bars indicating in frame and the others out of frames.

mRNA and RPF levels were at their highest throughout the viral genome (Fig. 2a).

To provide a general overview of our transcriptome, we quantified the amount of standard and nonstandard body transcription. Accordingly, 78.5% of the junctions were mapped to the canonical junctions which generate sgRNAs (Supplementary Fig. 2e), comparable to that of the previous report[11]. Also,

accumulation of mRNA-seq reads mapped to the upstream of TRS-L was observed in the positive strand, which is an expected feature of the SARS-CoV-2 transcriptome (Supplementary Fig. 2f)[11]. On the other hand, such accumulation was not observed in the negative strand (Supplementary Fig. 2f), due to the negligible amount of antisense transcription and translation, which corresponds to <0.1% of total mapped reads (orange bars

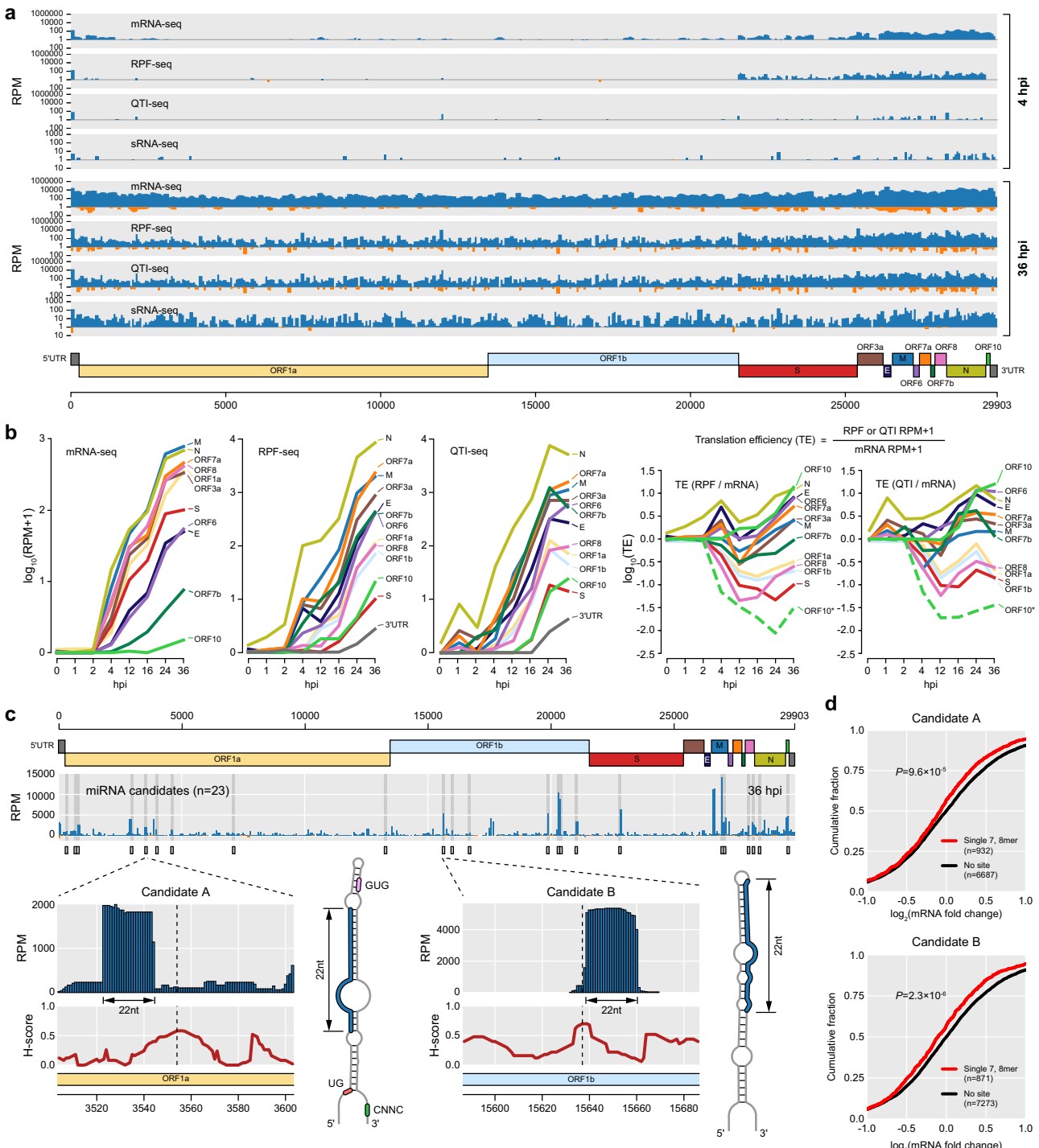

in Fig. 2a and Supplementary Fig. 2a–d, f–h). The detected antisense transcripts are likely to be remnants of antisense sgRNAs that were used as a template during viral transcription[20,21]. We also have analyzed our translatome data to observe programmed frameshifting (PRF) at ORF 1b, an important characteristic of coronavirus translation[13,22,23]. When measuring the PRF efficiency by comparing the read densities between ORFs 1a and 1b (Supplementary Fig. 2i), we obtained 64% efficiency on average (Supplementary Fig. 2j), comparable to a previous report[13]. Together, the consistency of our results firmly supports the high quality of our datasets.

When the temporal expression of individual SARS-CoV-2 genes was observed, the overall increment in expression level for all ORFs over time was observed on both mRNA and RPF levels (Fig. 2b). From 2 to 36 hpi, mRNA expression levels of ORFs M and N were highest followed by ORFs 7a, 8, 1a, 3a, S, 6, E, 7b, and 10. When translation of individual ORFs was observed using RPF-seq and QTI-seq, expression of ORF N was highest followed by ORFs 7a, M, and 3a (Fig. 2b). Interestingly, ORF 10 exhibited modest but clear level of translation during the late phase compared to 3′UTR, suggesting that ORF 10 is indeed translated and might be functional in contrast to a recent report[11]. By comparing the whole translatome and transcriptome, the translation efficiency (TE) of individual ORFs was examined (Fig. 2b). While ORFs 10, N, E, and 6 exhibited efficient TE, ORFs 1a, S, and 8 were not translated efficiently through the time

**Fig. 2 Temporal landscape of the SARS-CoV-2 translatome and transcriptome. a** Coverage of mRNA-seq, RPF-seq, QTI-seq, and sRNA-seq reads across the SARS-CoV-2 genome in early phase (4 hours post infection (hpi), top) and late phase (36 hpi, bottom) after viral infection (multiplicity of infection = 10). The y-axis represents the number of reads per million mapped reads (RPM) on a $log_{10}$ scale, and blue and orange bars indicate the number of reads mapped to positive and negative strands of the genome, respectively. **b** Expression level changes of SARS-CoV-2 ORFs over 0–36 hpi in the transcriptome (mRNA-seq, left) and translatome (RPF-seq and QTI-seq, middle) levels and translation efficiency (right). Expression levels of each ORF were measured as RPM for each time point (see "Methods") and displayed as $log_{10}(RPM + 1)$. Translation efficiency was calculated by dividing the translation level by the mRNA level for each ORF. ORF 10* indicates the translation efficiency of ORF 10 as the RPF expression level divided by mRNA expression level of N sgRNA (see "Methods"). **c** Two examples of potential miRNA candidates detected on the SARS-CoV-2 genome. sRNA-seq reads mapped on the SARS-CoV-2 genome are displayed as blue bars and the corresponding H-scores, which summarize the folding degree of the RNA hairpin structure centered on the nucleotide position (see "Methods"), are depicted as a red line below. Predicted RNA secondary structures of the two miRNA candidates are also illustrated where predicted mature miRNAs are shown in blue and known determinants for miRNA processing are indicated. **d** Repression of human mRNAs targeted by SARS-CoV-2 miRNA candidates identified in **c**. Expression fold changes of each mRNA after viral infection were measured by mRNA-seq for Calu-3 cells at 36 hpi. Human mRNAs containing a single 7, 8mer target site of the identified candidates in their 3′UTRs were selected (see "Methods). Cumulative distribution of $log_2$(mRNA fold change) of the target mRNAs was plotted (red) and compared with that of nontargets ("no site," black) by two-sided Wilcoxon's rank-sum test.

course (Fig. 2b). Although ORF S appears to be fairly weakly translated in this analysis, this is not the case after reflecting active TI from upstream leader sequence (see below). Our analysis demonstrates that the SARS-CoV-2 translatome is fairly dynamically regulated during viral infection, which is compelling and counterintuitive given that 5′UTRs have been known to include most regulatory elements for translation regulation[24,25] and that all the viral transcripts have identical 5′UTRs up until the TRS-L. Underlying molecular mechanisms for such dynamic TE regulation among the viral ORFs remain to be examined for a deeper understanding of the translational regulation of the SARS-CoV-2 translatome.

To identify potential SARS-CoV-2 miRNAs, we have performed sRNA-seq during the time-course experiment. Overall, the viral small RNA expression level increased after infection with the most abundant viral sRNA-seq reads detected at 36 hpi (Fig. 2a and Supplementary Fig. 2d) and sRNA expression was distributed fairly evenly throughout the entire viral genome. Using sRNA-seq data at 36 hpi, SARS-CoV-2 miRNA candidates were identified by selecting the regions that are highly expressed and where strong hairpins are formed (Fig. 2c and "Methods"). Several candidates, candidate A for example, exhibited miRNA-like stem-loop structures and primary sequence determinants known to be required for proper processing of mature miRNAs, such as basal UG, apical UGUG, or downstream CNNC motifs[26] (Fig. 2c). When investigating the human transcriptome response by the SARS-CoV-2 miRNA candidates, significant repression was observed from human mRNAs containing a single 7, 8mer target site of the identified candidates in their 3′UTRs (Fig. 2d). This result raises a compelling possibility of an interplay between the viral miRNAs and human gene expressions, and thus some of these miRNA candidates could play functional regulatory roles. The mechanism behind the biogenesis of SARS-CoV-2 miRNAs remains elusive since the viral transcription by the RNA-dependent RNA polymerase takes place in cytoplasm[9,27], while the initial processing of primary miRNAs takes place in the nucleus[28–30]. However, considering the previous reports that have reported the possibility of viral miRNA processing by Drosha translocated to cytoplasm upon viral infection[31,32], these miRNA candidates could also be processed by translocated Drosha in cytoplasm.

**TIS located in the leader (TIS-L).** Carefully dissecting the viral translatome, we noticed substantial amount of the RPF-seq and QTI-seq reads were mapped on a CUG codon located 10 nts upstream of TRS-L (Fig. 3a and Supplementary Fig. 3a). At 48 hpi, 22% of the total RPF-seq reads and 11% of QTI-seq reads

mapped to the SARS-CoV-2 genome originate from the CUG codon (Fig. 3b and Supplementary Fig. 3b), and similar results were consistently observed at different time points and MOIs (Supplementary Fig. 4a). It has been reported that eukaryotic translation can be initiated by noncanonical start codons (e.g., CUG, GUG, and UUG) as well as a canonical AUG codon[33,34], which generates alternative protein isoforms that potentially affect a variety of cellular functions[35–37]. Among them, CUG is known to be the most proficient TIS[33], and this noncanonical TIS located in the leader, which we termed TIS-L, is included in all gRNAs and sgRNAs of SARS-CoV-2 (Fig. 3a, b and Supplementary Fig. 3a, b). Enrichment of RPF-seq and QTI-seq reads at less proficient UUG codon at 5′ leader region had been identified in MHV and IBV[22,23]. A recent study has also detected CUG TIS-L in SARS-CoV-2[13], reporting that most of the TIS-L reads were mapped to gRNA. However, when focusing on a subset of RPF-seq reads of our dataset that initiate from TIS-L and that are long enough to be uniquely mapped to the SARS-CoV-2 genome, a vast majority of both RPF-seq and QTI-seq reads (>95%) obtained at 48 hpi were mapped to sgRNAs, while <5% of the reads were mapped to gRNA (Fig. 3c and Supplementary Fig. 3c, e). Similar results were consistently observed at different time points (16, 24, and 36 hpi) (Supplementary Fig. 4b, c). These results indicate that the TIS-L may be a global TIS for sgRNAs as well as gRNA with the largest number of TIS-L reads mapped to ORF 6, followed by ORFs N, 8, 7a, 1a, and S (Fig. 3d and Supplementary Fig. 3d).

Due to the short read length of the RPFs (~28 nt), the number of the RPF-seq and QTI-seq reads uniquely mapped to TIS-L accounts for a very small fraction. To overcome this problem, we relaxed RNase I concentration and repeated RPF-seq experiments. Even with this relaxed RNase I concentration, a vast majority of the uniquely mapped reads were mapped to sgRNAs (95%) and their relative distribution to SARS-CoV-2 ORFs was consistent with that of the high RNase I concentration (Fig. 3d, e and Supplementary Fig. 3f).

When comparing TI at annotated TISs with that of TIS-L based on RPF-seq and QTI-seq dataset obtained at 36 hpi, TI of TIS-L was even higher than that of the annotated TIS for several ORFs, including ORF S which encodes spike protein crucial for host recognition and viral entry[38] (Fig. 3f and Supplementary Fig. 4d). Since translation of ORF S was very low and almost absent near the annotated TIS, the RPF reads derived from TIS-L were added to correctly measure its translation level. When recalculating RPF expression level and TE for ORF S using this updated value, ORF S appears to be translated efficiently in contrast to our analysis shown in Fig. 2b (Fig. 3g and Supplementary Fig. 4e), suggesting that one should take TIS-L

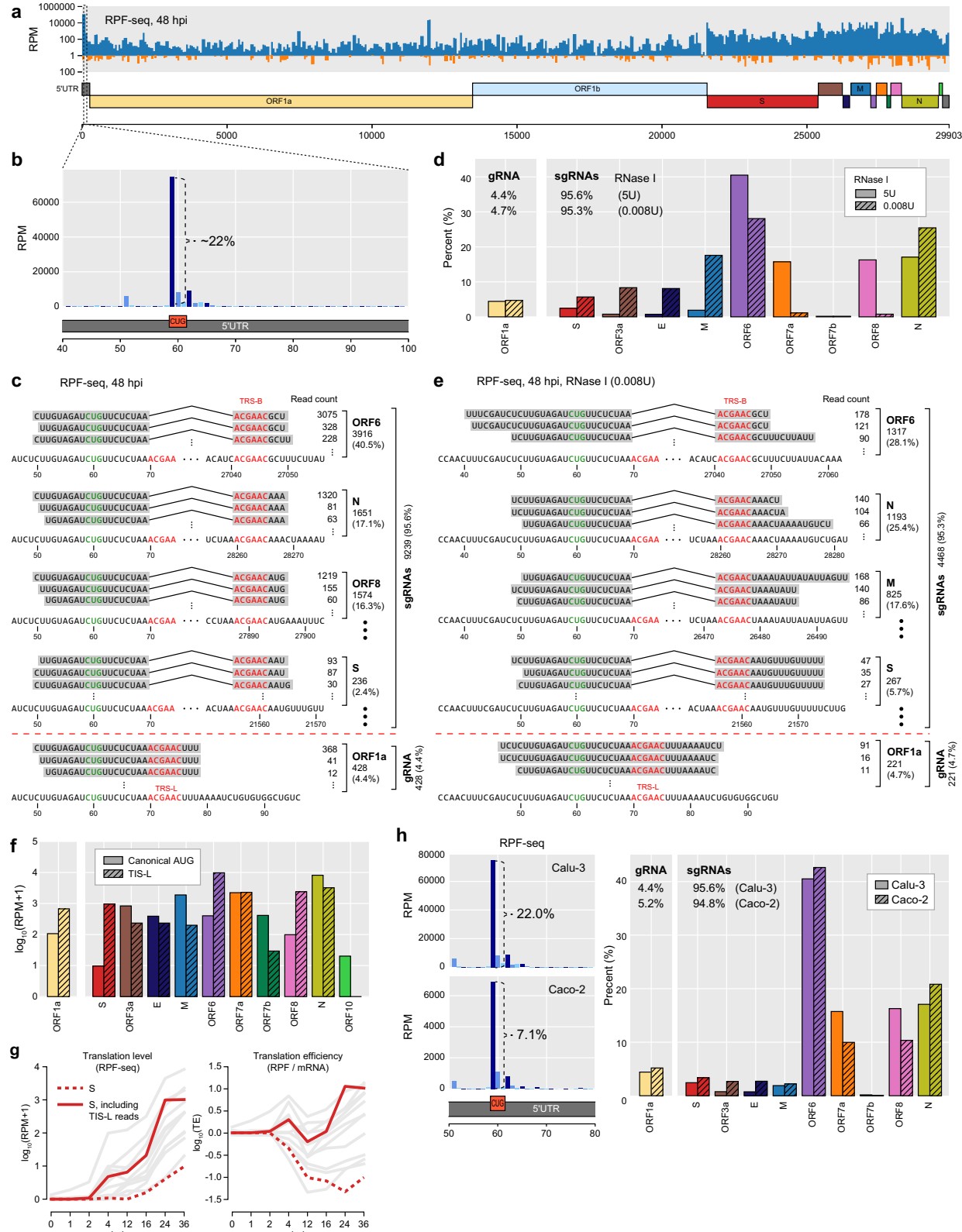

into careful consideration to more accurately quantify the SARS-CoV-2 translatome.

To examine whether our findings are general in other biological contexts, we have additionally performed RPF-seq and QTI-seq with Caco-2 cell line. Consistent with Calu-3, we observed that a considerable amount of reads were mapped to TIS-L in Caco-2 (Fig. 3h and Supplementary Fig. 6a). The relative

fraction of the TIS-L reads derived from each viral mRNA was also similar to that of Calu-3 (Fig. 3h and Supplementary Fig. 6a). Taken together, the observed enrichment of RPFs at TIS-L in another human cell line clearly demonstrates that this finding should be general enough to be applied to other cell types.

Interestingly, when same experiment was performed using Vero cells, although the TI at TIS-L was also observed, the

**Fig. 3 Extensive translation initiation by the translation initiation site located in the leader (TIS-L) for both gRNA and sgRNAs. a** Coverage of RPF-seq reads across the SARS-CoV-2 genome at 48 hours post infection (hpi) (multiplicity of infection = 0.1). Otherwise as in Fig. 2a. **b** Enrichment of RPF-seq reads at the TIS-L, which is a CUG codon located at 59th nt position in the leader sequence. The 13th position (12-nt offset from the 5′ end) of the reads, indicating the ribosome P-site position, was counted and calculated as the number of reads per million mapped reads (RPM). Open reading frames are depicted as three different colored bars with the dark blue bars indicating in frame with TIS-L and the others out of frames. **c** RPF-seq reads mapped to TIS-L categorized by whether their 3′ ends are mapped to the sgRNAs or the gRNA. For 48 hpi, a subset of RPF-seq reads around the TIS-L that were long enough to be uniquely mapped to the SARS-CoV-2 genome was collected (see "Methods"). The alignments of those uniquely mapped reads to the gRNA or sgRNAs are displayed and the corresponding read counts with the relative fraction of reads mapped to each ORF are shown in parentheses. **d** The relative fraction of TIS-L reads uniquely mapped to each ORF was compared between RPF-seq reads with high (5U, **c**) and low (0.008U, **e**) RNase I concentration (right). **e** An independent dataset of RPF-seq reads of TIS-L with reduced RNase I concentration (0.008U) at 48 hpi in Calu-3 cells, which consists of longer RPF-seq reads, were collected to obtain a larger number of reads uniquely mapped to TIS-L. Otherwise as in **c**. **f** For each ORF, level of translation initiation at TIS-L is compared with that at the annotated translation initiation site using RPF-seq dataset at 36 hpi. Otherwise as in **d**. **g** Using RPF-seq datasets, RPF expression level (left) and translation efficiency (right) of ORF S initiated from annotated ORF start codon (red dashed line) alone were measured and compared with those estimated by the number of RPF-seq reads from both annotated ORF start codon and TIS-L (see "Methods") (red, solid line). Translation levels or translation efficiencies of other ORFs are depicted with gray lines. Otherwise as in Fig. 2b. **h** Enrichment of RPF-seq reads at the TIS-L for Calu-3 and Caco-2 cell lines at 48 hpi (left). The relative fractions of TIS-L reads mapped to each of gRNA and sgRNAs are compared between Calu-3 and Caco-2 (right). Otherwise as in **b** and **d**.

ribosome occupancy at TIS-L was lower than in Calu-3 (Supplementary Fig. 6b). Furthermore, even though 93% of Vero TIS-L reads were consistently mapped to sgRNAs, the relative distribution to SARS-CoV-2 ORFs was remarkably different between two cell lines (Supplementary Figs. 4b and 6b, c). These discrepancies could have resulted from differences in cell line characteristics. Vero cells are African green monkey kidney cells that are widely used for viral replication since they are incapable of synthesizing IFNs[16,17]. Its lack of immune defense against viruses could be causing these inconsistencies between human cells and Vero and therefore viral and host translatome and transcriptome obtained from Calu-3 and Caco-2 may reflect a more physiologically relevant condition in human.

**TIS-L functions as a global regulator of the SARS-CoV-2 translatome**. Because TIS-L is located in the leader sequence, it should function as an upstream ORF (uORF) for each gRNA or sgRNA and the potential function of TIS-L is anticipated to differ for each ORF depending on its reading frame with respect to the annotated ORF (Fig. 4a). For instance, TIS-L is in frame with annotated ORF S and thus expected to act as an alternative TIS that can enhance the translation of ORF S (Fig. 4c and Supplementary Fig. 8b). Since S protein has a signal peptide at its N-terminal end that is cleaved during or after translocation to ER[39,40], mature S proteins translated from the annotated AUG and TIS-L are predicted to be identical after signal peptide cleavage[41] (Supplementary Fig. 7h). For ORFs 1a and N, TIS-L is expected to create a short uORF that is not overlapping with the annotated ORF (Fig. 4b and Supplementary Figs. 7f and 8a, j). Because TRS-B of ORF 6 is embedded in the middle of ORF M, TIS-L should produce an uORF that is overlapping and in frame with the C-terminal region of ORF M (Fig. 4d and Supplementary Fig. 8f). This uORF could produce a functional protein that consists of a cytoplasmic tail that is soluble due to the absence of the transmembrane domain of M protein[42].

In cap-dependent translation, translation at uORF would generally lead to the translation reduction of the downstream ORFs[43,44]. Most uORFs derived from TIS-L overlap with annotated ORFs and are out of frame with them (ORFs 3a, E, M, 7a, 7b, and 8), likely functioning as a translation suppressor of the annotated ORFs (Fig. 4e and Supplementary Figs. 7a–e and 8c–e, g–i). Since the TIS-L is an efficient TIS, a considerable number of ribosomes would recognize it and start the translation of a uORF. These ribosomes would stop at a stop codon of the uORF and cannot be used to translate the primary ORF, resulting

in the reduction of TI at primary ORF and a negative regulatory impact on its translation.

While it is known that ORF 7b is translated from the ribosomal leaky scanning of ORF 7a sgRNA[45], our data show that TIS-L produces an out-of-frame uORF for ORF 7b, which may not affect the ORF 7b translation by leaky scanning (Supplementary Figs. 7g and 8k). However, the functional role of putative ORF 7b sgRNA is unclear due to its low level of production (Fig. 2b) and inability to translate ORF 7b, and remains to be determined. ORF 10 does not have a clearly defined TRS-B at its 5′ end and is previously suggested to be not transcribed[11]. To help resolve the controversy over ORF 10 expression and function, we investigated the production of ORF 10 sgRNAs by noncanonical junctions in our mRNA-seq dataset and found that only <0.02% of the total junction-spanning reads occurred between TRS-L and the 5′ end of ORF 10. Moreover, most of these noncanonical junctions occurred downstream of ORF 10, and the hypothetical ORFs created from TIS-L were out of frame with ORF 10 (Supplementary Fig. 9a). However, TI at the annotated TIS, evident triplet nucleotide periodicity of the RPF-seq reads, and increased level of translation at 24 and 36 hpi were clearly detected for ORF 10 (Supplementary Fig. 9b–d), indicating that ORF 10 might produce a functional protein. These results collectively suggest that the translation of ORF 10 mainly occurs from sgRNAs of the other ORFs, rather than from its own sgRNAs, perhaps by ribosomal leaky scanning or reinitiation after the termination of translation for uORFs[46].

To evaluate the impact of TIS-L on SARS-CoV-2 translatome, we investigated a possible interplay between the uORF derived from TIS-L and translation of the annotated ORF using *Renilla* luciferase (RLuc) reporter system (Fig. 4f and Supplementary Fig. 9e and "Methods"). In all tested cases (ORFs S, 6, and 7a), the replacement of CUG with CCG marginally affected the relative RLuc expression under uninfected conditions (Fig. 4f and Supplementary Fig. 9e). By contrast, the uORF derived from TIS-L positively or negatively affected the relative RLuc expression under SARS-CoV-2-infected conditions depending on its reading frame with respect to the annotated ORF. In case of S, CUG promoted the relative RLuc expression by ~2.6-fold compared to the CCG. For ORF 6, uORF derived from TIS-L that is not overlapping with downstream ORF had no significant effect on RLuc expression. On the other hand, the presence of CUG in ORF 7a mRNA reduced the amount of RLuc expression by ~2-fold, compared to the CCG, suggesting that this uORF derived from TIS-L functions as a translation suppressor for ORF 7a expression. In all cases, CUG/CCG did not significantly affect the

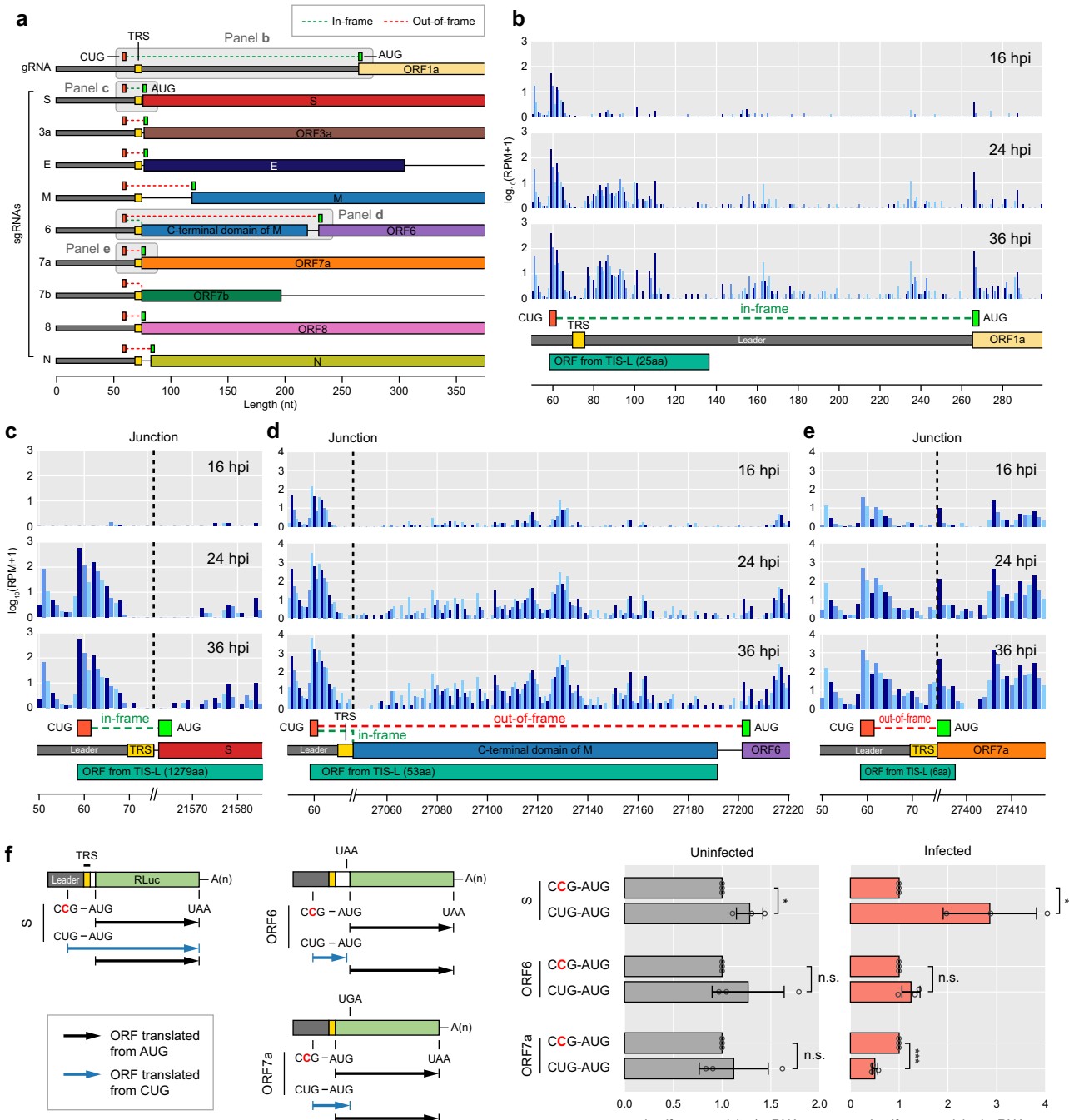

**Fig. 4 Translation initiation site located in the leader (TIS-L) functions as a global regulator of the SARS-CoV-2 translatome. a** Overview on the position of TIS-L in relation to ORF start codons of SARS-CoV-2 gRNA and sgRNAs. For each of gRNA and sgRNAs, positions of TIS-L (orange), transcription regulatory sequence (TRS, yellow), and start codon of ORFs (green) were displayed. The reading frame of each TIS-L-initiating hypothetical ORF was shown as a green (in frame) or red (out of frame) dashed line compared to the reading frame of the annotated ORF. RPF-seq reads mapped on TIS-L compared to those of the annotated ORFs 1a (**b**), S (**c**), 6 (**d**), and 7a (**e**), measured at 16, 24, and 36 hpi with their nucleotide sequence, annotation of each ORF, TRS, TIS-L, and a predicted ORF initiated from TIS-L shown below. For reading frames, the dark blue and the other blue bars indicate RPF-seq reads that are in frame and out of frame with the annotated ORF, respectively. The *x*-axis represents SARS-CoV-2 genomic position, and the *y*-axis represents $\log_{10}(RPM + 1)$. The dashed lines indicate junction positions of each sgRNA. The ORFs that starts from CUG or TIS-L are depicted with its calculated protein sizes. Whether the CUG-derived hypothetical ORFs are in frame or out of frame with respect to the annotated downstream ORFs is also shown for each viral ORF. RPF-seq and QTI-seq reads mapped on the other ORFs are shown in Supplementary Figs. 7 and 8. **f** Experimental validation of TIS-L functions. A schematic diagram of RLuc reporters are displayed with the mutated sequences shown in red, designed to disrupt TIS-L-initiated uORF. Calu-3 transiently transfected with a RLuc reporter and a plasmid expressing FLuc mRNA were either infected or uninfected with SARS-CoV-2 and the relative RLuc activities were measured. RLuc and FLuc activities were normalized to RLuc and FLuc mRNAs, respectively (two-tailed, equal-sample variance Student's *t* tests, *P < 0.05, **P < 0.01, ***P < 0.001). The mean values ± s.d. are displayed (*n* = 3 biologically independent experiments). *P* values are provided in Source Data.

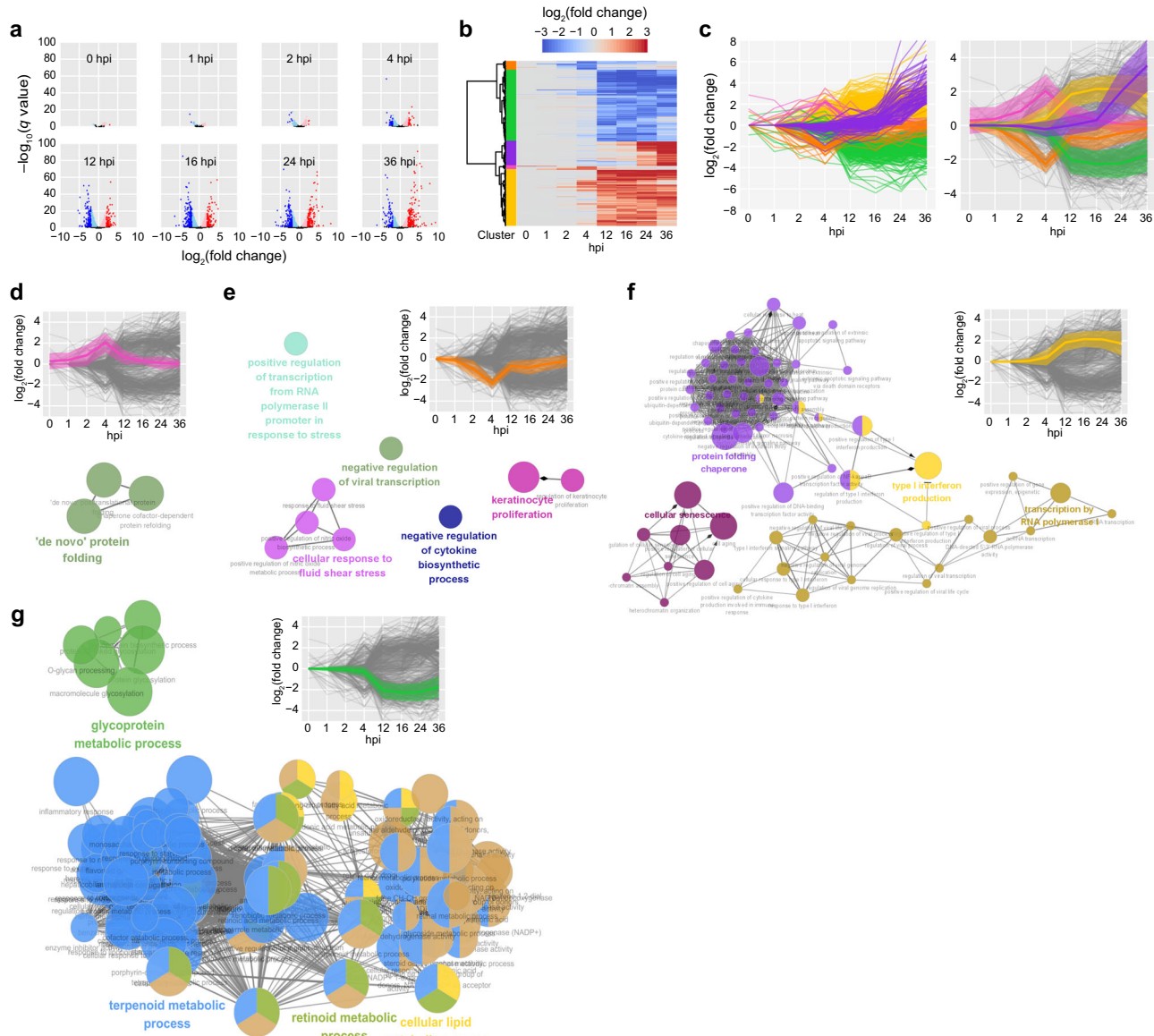

**Fig. 5 Early and late responding human genes to SARS-CoV-2 infection. a** Differentially expressed genes (DEGs) identified from RPF-seq data are shown for each time point. From 0 to 36 hpi, RPF-seq levels of human genes were compared with those of uninfected condition. Log$_2$(expression fold change) ($x$-axis) and statistical significance ($y$-axis; $-\log_{10}$-scale FDR-corrected $q$ value) of each gene are displayed (see "Methods"). For a subset of DEGs with $q < 0.01$, highly upregulated and downregulated DEGs with $|\log_2(\text{expression fold change})| \geq 2.0$ are indicated in red and blue, respectively, and moderately upregulated and downregulated DEGs with $|\log_2(\text{expression fold change})| < 2.0$ are indicated in pink and light blue, respectively. **b** Hierarchical clustering of the DEGs displayed in **a**. The genes identified as DEGs at least for one time point were clustered with respect to their temporal expression patterns (see "Methods"). Upregulation and downregulation in comparison to the uninfected condition is indicated in red and blue, respectively. **c** Temporal expression changes of the identified DEGs color-coded for the five clusters determined in **b** (left). Overall expression changes for each cluster were also depicted by mean log$_2$(expression fold changes) (right). The light-colored shades for each line indicate the standard deviations, and the gray lines represent the expression changes of individual DEGs (left). For the DEGs included in each cluster determined in **b**, Gene Ontology (GO) enrichment analysis were performed and GO terms associated with early (**d**, **e**) and late (**f**, **g**) responding clusters in response to the viral infection are shown. For each cluster, top five GO terms chosen based on statistical significance and their temporal expression patterns are displayed (see "Methods").

levels of mRNAs. Together with our observation of the high level of TI at TIS-L (Fig. 3f), we postulate that the regulatory impact of TIS-L on the SARS-CoV-2 translatome may play critical roles for SARS-CoV-2 pathophysiology.

**The impact of SARS-CoV-2 on the human translatome and transcriptome.** To assess the impact of viral infection on human translatome and transcriptome, we have identified differentially expressed genes (DEGs) for each time point from RPF-seq

(Fig. 5a). As the time after infection elapses, the number of DEGs increased (Fig. 5a). These DEGs were clustered based on their temporal expression patterns (Fig. 5b). Each cluster exhibited a unique temporal expression pattern and some clusters showed significantly different expression patterns between early (0, 1, 2, and 4 hpi) and late (12, 16, 24, and 36 hpi) time points (Fig. 5c). Gene Ontology (GO) analysis further showed various host responses to viral invasion during early and late phases (Fig. 5d–g and Supplementary Fig. 10d). During the early phase, genes that are involved in protein folding were upregulated, while genes that

negatively regulate viral transcription were downregulated (Fig. 5d, e). Of note, heat shock protein production is known to be upregulated during cellular stress[47].

During late time post infection, upregulation of type I IFN production was observed (Fig. 5f). Intriguingly, genes involved in the response to cellular hormone metabolic process or retinoic acid metabolic process were downregulated (Fig. 5g). Steroid hormones are known to regulate various biological processes, including metabolism, inflammation, and immune functions[48]. Therefore, the observed downregulation and translational repression of the genes implicated in steroid hormone response would be advantageous for SARS-CoV-2 survival. Supporting this hypothesis, recent studies showed that a treatment with dexamethasone, a potent chemical derivative of glucocorticoid, mitigates the damaging immune response to SARS-CoV-2[49–51]. Similarly, DEGs were identified and GO analysis was performed for mRNA-seq and QTI-seq data (Supplementary Figs. 10a–c and 11a–e) with the overall results consistent with that of RPF-seq.

**Functions and pathways of human genes responding to SARS-CoV-2 infection**. SARS-CoV-2 has been known to interact with angiotensin-converting enzyme 2 (ACE2) receptor and TMPRSS2, which are essential for viral entry and thus targeted by various currently developing therapeutic strategies[52]. ACE2 receptor binds to SARS-CoV-2 S protein and TMPRSS2 is a serine protease that primes and activates S protein to trigger membrane fusion, enabling viral entry into the host cell[38,53]. When examining the temporal expression pattern of ACE2, the level of *ACE2* mRNA did not change over time and yet its translation level decreased gradually over time (Fig. 6a). This observation is consistent with previous reports that ACE2 expression decreases upon SARS-CoV infection and this reduction worsens the acute respiratory distress syndrome[52,54,55]. Our analysis delineates that downregulation of ACE2 induced by viral entry takes place on a translation level. On the other hand, TMPRSS2 was downregulated during early phase and then upregulated during late phase, exhibiting a similar expression pattern between mRNA-seq and RPF-seq, which suggests the gene expression is mainly regulated transcriptionally and/or posttranscriptionally (Fig. 6a). When the translatome and transcriptome of the 332 proteins reported to be interacting with SARS-CoV-2 proteins were examined, no drastic changes were observed compared to the other genes[56]. However, when investigating the temporal expression patterns of host factors critical for SARS-CoV-2 infection[57], the differential expression was identified for multiple host factors including the targets of existing drugs (Fig. 6b). Intriguingly, the host factors with larger changes in their expressions upon SARS-CoV-2 infection tend to be more critical for SARS-CoV-2 infectivity (Fig. 6c), demonstrating how our dataset can be utilized to select more effective therapeutic targets.

IFN signaling is recognized as the first line of defense as it is the integral component in innate immune response[58]. SARS-CoV-2 restricts the production of host type I and III IFN while worsening inflammation by increasing cytokines and chemokines[59,60]. However, a recent study showed that when cells are infected at high MOI, the ability of SARS-Cov-2 to inhibit IFN-I and III production is restricted[59]. Consistent with this result, our mRNA-seq, RPF-seq, and QTI-seq data showed the upregulation of IFNB1, IFNL1, IFNL2, IFNL3, as well as IFN-stimulated genes during the late phase (Fig. 6d). On the other hand, while a number of cytokines have been previously reported to be upregulated upon SARS-CoV-2 infection[59], only a few of them exhibited consistent upregulation and the others were either unchanged or even downregulated in our analysis (Fig. 6e).

To identify GO terms and pathways associated with DEGs detected at each time point by mRNA-seq, RPF-seq, and QTI-seq, we have performed GO enrichment and DAVID Kyoto Encyclopedia of Genes and Genomes (KEGG) pathway analyses (Fig. 6f, g and Supplementary Fig. 11f, g). While only a small number of GO terms were marginally significantly associated with the DEGs detected during early phase, a large number of GO terms and pathways that are related to metabolite glucuronidation, type I IFN signaling pathways, and viral infection were significantly associated with the DEGs detected during late phase (12, 16, 24, and 36 hpi) (Fig. 6f, g and Supplementary Fig. 11f–h). Finally, using our sRNA-seq dataset, miRNA expression levels at each time point were compared to detect differentially expressed host miRNAs (Fig. 6h). From 0 to 36 hpi, three miRNAs (mir-96, 100, and 196a) were gradually upregulated, while four miRNAs (mir-7, 23b, 193b, and 625) were downregulated after infection (Fig. 6h). Their viral and host targets remain to be determined and will be interesting subject of future studies.

**Discussion**
We have constructed massive-scale datasets of mRNA-seq, RPF-seq, QTI-seq, and sRNA-seq at various time points (0-48 h) in order to observe the translatome and transcriptome dynamics upon SARS-CoV-2 infection (Fig. 1a). Strikingly, a large fraction (11–22%) of SARS-CoV-2 RPF-seq and QTI-seq reads were mapped to a TIS-L that is a noncanonical CUG codon (Fig. 3b and Supplementary Fig. 3b) and a vast majority (>95%) of the TIS-L reads were mapped to sgRNAs (Fig. 3d and Supplementary Fig. 4a–c). These results were consistently observed at various time points and for relaxed RNase I conditions (Fig. 3d, e), suggesting that translation regulation by TIS-L may be widespread in the SARS-CoV-2 translatome.

TIS-L is anticipated to function as uORFs (Fig. 4 and Supplementary Figs. 7 and 8) with a different functional consequence for each gRNA or sgRNA depending on its reading frame with respect to the annotated ORF. These uORFs can be categorized as follows: (i) in-frame uORF overlapping, (ii) out-of-frame uORF overlapping, and (iii) uORF nonoverlapping with the annotated ORF. We have performed luciferase assays to examine the function of TIS-L using ORFs S, 7a, and 6 for categories (i), (ii), and (iii), respectively (Fig. 4f and Supplementary Fig. 9e). Accordingly, we discovered that TIS-L for ORF S that makes an in-frame uORF overlapping with the annotated ORF acts as a strong translation enhancer by providing an alternative TIS for ORF S. On the other hand, TIS-L for ORF 7a that makes an out-of-frame uORF overlapping with the annotated ORF negatively regulates translation (Fig. 4f and Supplementary Fig. 9e) and this is the way by which TIS-L functions for most of the other SARS-CoV-2 ORFs. In case of a rather exceptional ORF 6 where TIS-L creates an uORF nonoverlapping with the annotated ORF, TIS-L did not affect its translation (Fig. 4f and Supplementary Fig. 9e). Taken together, these results indicate the impact of TIS-L on the SARS-CoV-2 translatome is widespread and therefore TIS-L may function as a global regulator of the SARS-CoV-2 translatome, although the detailed mechanisms of how TIS-L acts as the global regulator require further assessment.

Since several TI factors affected the rate of the TI at the non-canonical start codons[33], we have examined the temporal expression patterns of these TI factors upon SARS-CoV-2 infection. Accordingly, eIF5 was particularly upregulated in the late phase after viral infection (Fig. 6i), consistent with the previous reports that the increase of eIF5 level promotes non-canonical TI[61,62]. We postulate that eIF5 may be a key TI factor for the active TI at TIS-L, leading to a more severe pathophysiological consequence compared to other betacoronaviruses.

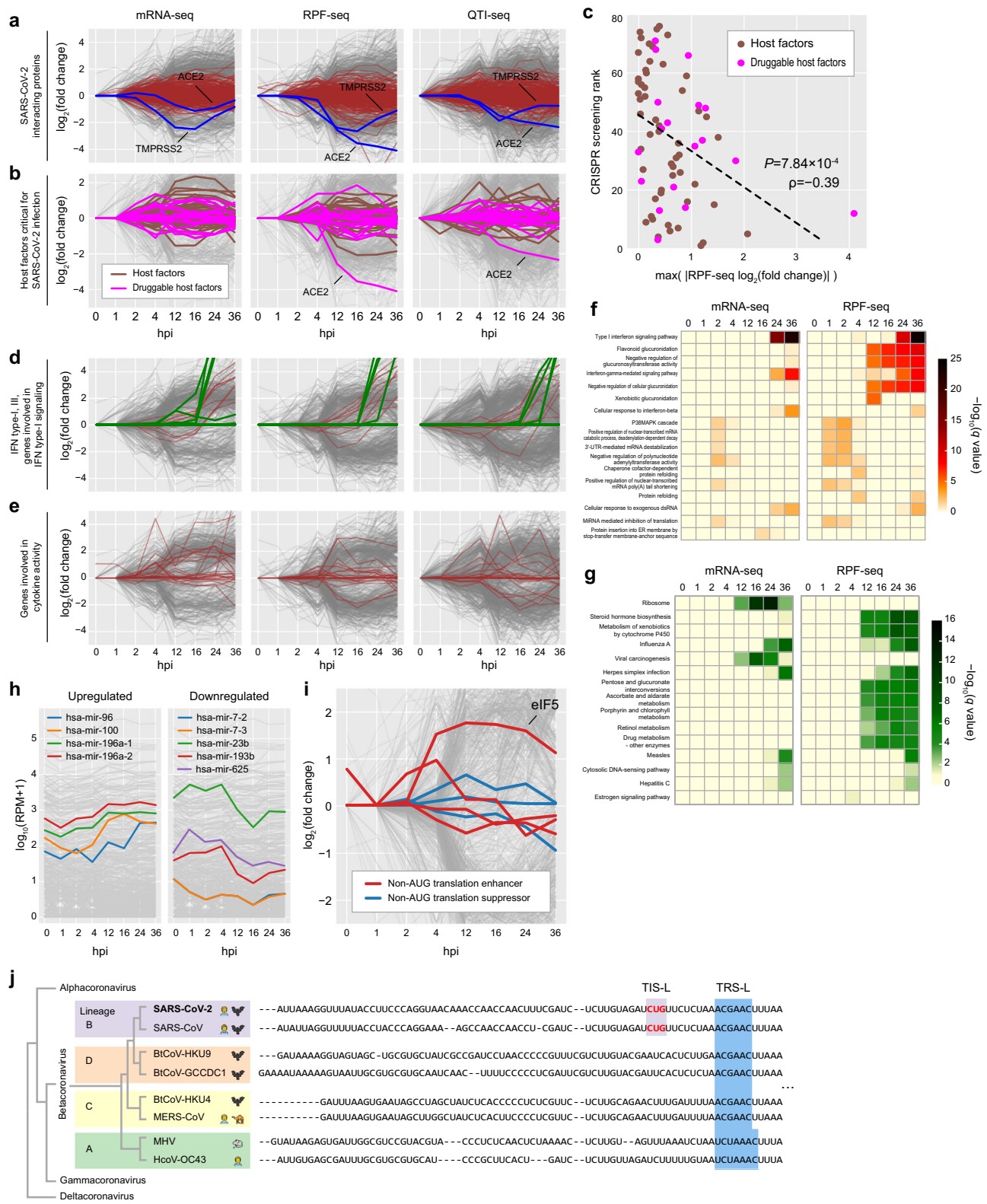

To gain an evolutionary insight into TIS-L in betacoronaviruses, we examined whether other betacoronavirus stains also have TIS-L in their leader sequences. We chose two representative betacoronaviruses for each of A, B, C, and D lineages and compared their leader sequences (Fig. 6j and see also Supplementary Fig. 11i for a full list). Interestingly, while all of the eight betacoronaviruses contained a UUG codon in their leader

sequences, which has been reported to be functional in MHV and IBV[22,23], only SARS-CoV and SARS-CoV-2 contained a CUG codon upstream of TRS-L. It has been known that the global translation of eukaryotic cells is suppressed while noncanonical translation is induced upon cellular stress such as viral infection[33] and that a CUG codon is most efficient noncanonical codon for TI[33]. Perhaps, when the lineage B of betacoronaviruses emerged

**Fig. 6 Associated functions and pathways of human genes responding to SARS-CoV-2 infection. a** Temporal expression changes of the human genes whose protein products were detected to interact with SARS-CoV-2 proteins[56] (brown) with ACE2 and TMPRSS2 highlighted in blue. For mRNA-seq (left), RPF-seq (middle), and QTI-seq (right) data, log$_2$(expression fold changes) for the genes at each time point were measured. Otherwise as in Fig. 5c. **b** Temporal expression changes of the host factors required for SARS-CoV-2 infection (brown), identified by CRISPR screening[57]. Host factors whose targeting drugs were found in the Drug Gene Interaction database (DGIdb) are highlighted in magenta. The gray lines represent the expression changes of individual differentially expressed genes (DEGs) identified in Fig. 5. For mRNA-seq (left), RPF-seq (middle), and QTI-seq (right) data, log$_2$(expression fold changes) for the genes at each time point were measured. **c** Association between the magnitude of differential expression and the impact on SARS-CoV-2 infectivity for the host factors investigated in **b**. The x-axis represents the maximum of the absolute values of RPF-seq log$_2$(expression fold changes) across the time points. The y-axis represents the ranks of the CRISPR screening enrichment for the host factors[57]. Spearman's correlation $\rho$ was measured between the two values, and the P value was obtained by using two-sided Student's t test. Otherwise as in **b**. **d** Temporal expression changes of the type I and III interferon (IFN-α, β, ε, κ, λ, and ω) (green) and previously reported DEGs involved in type I interferon response[59] (brown). Otherwise as in **a**. **e** Temporal expression changes of the previously reported DEGs involved in cytokine and chemokine activities[59] (brown). Otherwise as in **a**. **f** Gene Ontology (GO) terms associated with DEGs identified from mRNA-seq (left) and RPF-seq (right) data are shown from 0 to 36 hpi. At each time point, statistical significance of the GO terms is visualized as a heat map, color-coded by −log$_{10}$(FDR-corrected q values). **g** DAVID KEGG pathways associated with DEGs identified from mRNA-seq (left) and RPF-seq (right) data are shown from 0 to 36 hpi. Association significance of each KEGG pathway with mRNA-seq data (left) and RPF-seq data (right) is shown. Otherwise as in **f**. **h** Expression levels of human miRNAs upon SARS-CoV-2 infection at each time point with upregulated (left) and downregulated (right) miRNAs highlighted. Expression levels of each miRNA were measured as RPM for each time point (see "Methods") and displayed as log$_{10}$(RPM + 1). **i** Temporal expression changes of the genes reported to enhance (red) or suppress (blue) the translation initiation at non-AUG codons[33]. Otherwise as in **a**. **j** Multiple sequence alignments for the 5′ leader region of betacoronaviruses. Two representative viruses from each lineage are displayed. See Supplementary Fig. 11i for multiple sequence alignments for a full list of betacoronaviruses.

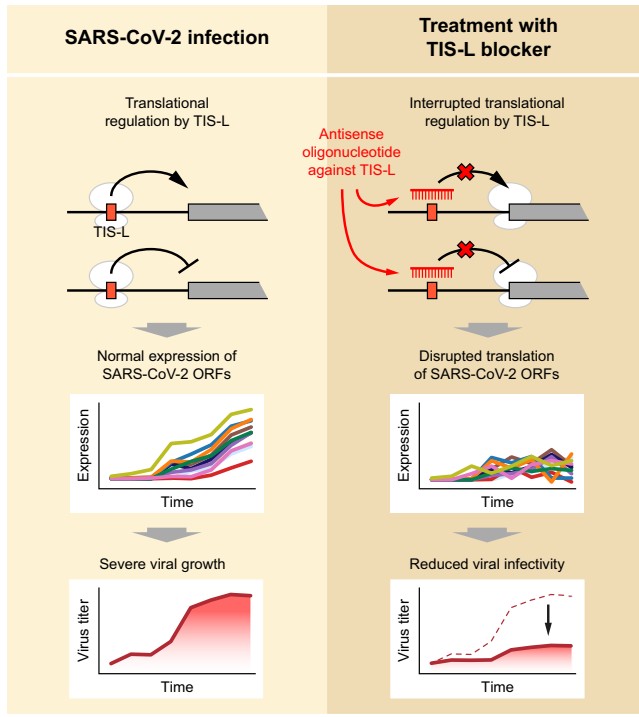

**Fig. 7 A hypothetical therapeutic strategy for SARS-CoV-2 treatment by blocking TIS-L with antisense oligonucleotide (ASO).** By blocking the TIS-L with ASO, the viral infectivity of SARS-CoV-2 could be reduced by the disruption of the translation of the viral ORFs.

in the evolutionary history, a mutation that produced a CUG codon in the leader region may have conferred a strong selective advantage by creating an efficient TIS and thus providing a competent strategy that bypasses the reduced global translation of the host cells in response to viral infection (Supplementary Fig. 11i). Functional roles of TIS-L in viral pathogenicity of the SARS-CoV lineage and its physiological consequences remain to be determined.

TIS-L-mediated regulation of the SARS-CoV-2 translatome may be applied to develop a novel therapeutic strategy. Since TIS-L appears to act as a master regulator of the SARS-CoV-2 translatome, blocking this region with an antisense oligonucleotide might

lead to efficient therapeutics by disrupting the TIS-L-mediated translational regulation of SARS-CoV-2 (Fig. 7). Our high-resolution temporal atlas of the translatome and transcriptome of the SARS-CoV-2 and its human host will serve as a useful resource to reveal the molecular basis of the SARS-CoV-2 pathogenicity and to urgently discover effective therapeutic strategies.

## Methods

**Cell lines and virus.** Calu-3 (KCLB Cat# 30055), Caco-2 (KCLB Cat# 30037.1), and Vero cells (KCLB Cat# 10081) were purchased from the Korean Cell Line Bank (Seoul, Korea). SARS-CoV-2 (BetaCoV/Korea/KCDC03/2020, NCCP no. 43326) was obtained from the Korea Disease Control and Prevention Agency (Osong, Korea).

The cells were cultured in the appropriate medium (Calu-3 and Vero cells: Dulbecco's modified Eagle medium (DMEM; Gibco, Waltham, MA, USA); Caco-2 cell: minimum essential medium Eagle with Earle's BSS (Lonza, Basel, Switzerland)), supplemented with 10% fetal bovine serum (FBS; Serana, Pessin, Germany) and 1% penicillin/streptomycin (Gibco), and maintained in a 5% $CO_2$ incubator at 37 °C. SARS-CoV-2 (BetaCoV/Korea/KCDC03/2020; GISAID accession ID: EPI_ISL_407193) was propagated in Vero cell and viral titers were determined by plaque assay as described below.

**Growth kinetics of SARS-CoV-2 in Calu-3 and Vero cells.** Calu-3 and Vero cells were seeded in 100 mm dishes, washed twice with phosphate-buffered saline (PBS; Lonza), and infected with SARS-CoV-2 at a MOI of 10 or 0.1 for 1 h. Next, the viral inoculum was removed, and infected cells were washed thrice with PBS and then maintained in DMEM supplemented with 2% FBS (Serana) and 1% penicillin/streptomycin (Gibco) in a 5% CO2 incubator at 37 °C. The supernatant was collected at 0, 1, 2, 4, 12, 16, 24, and 36 hpi, and titration was performed by plaque assay in Vero cells. All SARS-CoV-2-related experiments were performed at a BSL3 facility at Korea University (Seoul, Korea).

**Plaque assay.** Vero cells were seeded in 6-well dishes and inoculated with the SARS-CoV-2-infected supernatant for 1 h. Next, the viral inoculum was removed, and the cells were covered by Dulbecco's modified Eagle medium/nutrient mixture F-12 overlay medium (Sigma-Aldrich, St. Louis, MO, USA) containing 0.2% agarose (Oxoid). After 3 days, the cells were stained with crystal violet (Georgia Chemicals Inc.).

**Library preparation.** For ribosome-protected mRNA fragment sequencing (RPF-seq), QTI-seq, mRNA-seq, and sRNA-seq library construction, Calu-3, Caco-2, and Vero cells were seeded in 100 mm dishes and were infected with SARS-CoV-2 (either MOI 0.1 or 10) for indicated time points following the protocol above. Calu-3 cells were lysed at 0, 1, 2, 4, 12, 16, 24, 36, and 48 hpi, Caco-2 cells were lysed at 48 hpi, and Vero cells were lysed at 0 and 24 hpi as described below. Calu-3 and Vero cells were preincubated with either 100 μg/ml CHX (MilliporeSigma) or 2 μg/ml Harr (LKT Laboratories) for 10 min. After incubation, the cells were washed with cold PBS containing 100 μg/ml CHX and lysed in 600 μl of lysis buffer [10 mM Tris-HCl (pH 7.4), 5 mM MgCl2, 100 mM KCl, 1% Triton X-100, 100 μg/

ml CHX, 1 mM dithiothreitol, 0.2 U/µl RiboLock RNase Inhibitor (Thermo Fisher Scientific), and 1×EDTA-free protease inhibitor cocktail (Roche)] adapted from Bartel lab's protocol[19], followed by incubation for 10 min on ice. For QTI-seq with Caco-2 cells, the lysis protocol was modified by increasing the 2 µg/ml Harr (LKT Laboratories), preincubating from 10 to 20 min, and including 2 µg/ml Harr to the lysis buffer. After centrifugation at $10,000 \times g$ for 10 min at 4 °C, the supernatant was equally divided into two tubes for mRNA-seq and either RPF-seq or QTI-seq.

For RPF and QTI-seq library construction, 300 µl of the supernatant was digested with 5 U/µl RNase I (Ambion) for 45 min at room temperature. The RNase I-treated samples were loaded onto Illustra MicroSpin S-400 HR Columns (GE Healthcare) and centrifuged at $600 \times g$ for 2 min. The collected RPFs were subjected to ribosomal RNA depletion using RiboMinus™ Eukaryote System v2 (Thermo Fisher Scientific). To remove the phosphate groups at the ends of RPFs, the RPFs were treated with Antarctic phosphatase (NEB) at 37 °C for 1 h, and then the enzyme was heat inactivated for 5 min at 65 °C. For the phosphorylation of the 5′ end of RPFs, the RPFs were incubated with T4 polynucleotide kinase (NEB) in the presence of 1 mM ATP (Invitrogen) for 1 h at 37 °C, followed by gel purification for size selection using 12% TBE-Urea gel system (National Diagnostics). The gel slices containing RNA species corresponding to 26–32 nts were obtained and eluted with 400 µl of 0.3 M NaCl for overnight at 4 °C. To remove the gel debris, the eluted RPFs were subjected to a purification with Spin-X column (Corning). The purified RPFs were obtained by ethanol precipitation. Finally, the RPFs were subjected to library construction using TruSeq Small RNA Library Preparation Kits (Illumina) according to the manufacturer's protocol.

For mRNA-seq library construction, the total RNA was extracted from the aliquoted lysate using Trizol following the manufacturer's protocol (Qiagen). Then, total RNA was either treated with Ribo-Zero rRNA Removal Kit (Illumina) or RiboMinus™ Eukaryote System v2 (Thermo Fisher Scientific) to remove rRNA. The isolated mRNA was fragmented for 2.5 min using fragmentation reagents (Invitrogen) and cleaned up using RNeasy MinElute Cleanup Kit (Qiagen). The fragmented RNA was treated with phosphatase and PNK as described above and 200-400 nt size fragments were isolated by gel election. The rest of the library was prepared as described above using TruSeq Small RNA Library Preparation Kits (Illumina).

For sRNA-seq library, cells were lysed directly from dishes using Trizol and small RNA was isolated using Purelink™ miRNA isolation Kit (Invitrogen) kit. Then library was prepared using TruSeq Small RNA Library Preparation Kits (Illumina) according to the manufacturer's protocol.

**Preprocessing and alignment of sequenced reads**. RPF-seq, QTI-seq, mRNA-seq, and sRNA-seq libraries were sequenced using HiSeq 2500 (single-end, 100 bp for mRNA-seq and 51 bp for the rest; Illumina). Sequenced reads were pre-processed through four steps as listed below. Low sequencing quality regions were trimmed with in-house FASTQ quality trimming software, and adapter sequences (Illumina TruSeq RA3) at the 3′ ends were removed by cutadapt-1.2.1[63]. Sequencing artifacts or PCR duplicates were eliminated by fastx_artifact_filter of the FASTX-Toolkit package (hannonlab.cshl.edu/fastx_toolkit/). Reads derived from noncoding RNAs (ncRNAs) including tRNA and rRNA genes were removed by aligning to the ncRNA sequences using Bowtie2[64] with default parameters. For RPF-Seq, QTI-seq, and mRNA-seq, reads derived from miRNA genes were additionally discarded. We also removed reads from mycoplasma transcripts by aligning to the most common 4 mycoplasma genomes (*M. hominis*, *M. hyorhinis*, *M. fermentans*, and *A. laidlawii*) (Supplementary Table 1).

For RPF-seq, QTI-seq, and mRNA-seq, preprocessed reads were aligned to the SARS-CoV-2 genome (NC_045512.2) and human genome (hg38) by STAR aligner 2.7.5b[65]. For an alignment to the SARS-CoV-2 genome, we used the following parameters: "--outStd BAM_SortedByCoordinate --outReadsUnmapped Fastx --outSAMtype BAM SortedByCoordinate --outSAMattributes Standard --outFilter Type BySJout --outFilterMismatchNoverReadLmax 0.04 --chimOutType Within BAM HardClip --alignEndsType EndToEnd --outSJfilterOverhangMin 12 12 12 12 --outSJfilterCountUniqueMin 1 1 1 1 --outSJfilterCountTotalMin 1 1 1 1 --outSJfilterDistToOtherSJmin 0 0 0 0 --scoreGapNoncan 0 --scoreGapGCAG 0 --scoreGapATAC 0 --alignSJstitchMismatchNmax -1 -1 -1 -1 --alignSJoverhang Min 6 --alignSJDBoverhangMin 6." For RPF-seq and QTI-seq samples, we used the option "--alignSJDBoverhangMin 1" for higher sensitivity instead of "--alignSJD BoverhangMin 6".

For an alignment of host genome (hg38 for human and chlSab2 for green monkey), we collected unmapped reads from the alignment step of SARS-CoV-2 and aligned them to host genome. We used following parameters: "--outStd BAM_SortedByCoordinate --outReadsUnmapped Fastx --outSAMtype BAM SortedByCoordinate --outSAMattributes Standard --outFilterType BySJout --outFilterMismatchNoverReadLmax 0.04 --chimOutType WithinBAM HardClip --alignEndsType EndToEnd --alignSJoverhangMin 6 --alignSJDBoverhangMin 6".

For sRNA-seq, preprocessed reads were aligned to the SARS-CoV-2 genome and reference human genome (hg38) with Bowtie 2 with following parameters: "-t -k 101 --very-sensitive --norc --score-min C,0,0 --mp 8,8 --np 8." Unmapped reads were aligned again using Bowtie 2 with a more lenient score cutoff "-t -k 101 --very-sensitive --norc --score-min C,-8,0 --mp 8,8 --np 8," which allows a single mismatch during alignment.

**Representative isoforms of human mRNAs**. We obtained the UCSC refFlat annotation (hg38) curated from NCBI's RefSeq database for human genes[66] to collect a reference mRNA set. To obtain a single representative isoform from multiple mRNA isoforms for each gene, a series of filtering steps adapted from a previous study[67] were applied. Briefly, we collected NM sequences to take only mRNAs, eliminating ncRNAs. mRNAs mapped to the 24 human chromosomes were included while discarding others including "random" sequences. mRNA isoforms that contain a wrong start or stop codon or with lengths that are not a multiple of three were discarded. Candidates for nonsense-mediated mRNA decay, whose stop codon is located >50 nts upstream of the last exon-exon junction[68], were also filtered out. Accordingly, we obtained 18,997 representative non-redundant isoforms and used this set of reference mRNAs for subsequent computational analyses.

**Analysis for global overview of transcriptome and translatome dataset**. For each time point and each cell line, the length distributions for RPF-seq and QTI-seq reads mapped to the human or SARS-CoV-2 genome were depicted as a histogram (Supplementary Fig. 1d). To investigate whether the triplet periodicity of the mapped position of reads is observed for RPF-seq and QTI-seq data, 13th nt positions of the RPF-seq or QTI-seq reads within human or SARS-CoV-2 ORFs were counted. The relative fractions of the reads were obtained with respect to the frame of annotated ORFs (Supplementary Fig. 1e). To provide a global overview for the distribution of reads near the start of ORFs, human or SARS-CoV-2 genes with >50 reads mapped to ORF were collected. For each ORF, 13th nt positions of the reads were counted, and the number of the reads for each position was normalized by the amount of reads mapped to the entire ORF. The average of the relative fractions for the collected ORFs was displayed (Fig. 1e). Efficiency of programmed ribosomal frameshifting was measured by comparing the numbers of reads mapped to ORFs 1a and 1b, which were normalized by the length of each ORF (Supplementary Fig. 2j). In this analysis, those reads mapped to the region where ORFs 1a and 1b overlap to each other were excluded.

**Read coverage plots of sequencing data**. Sequencing read coverage was calculated using BEDtools genomecov (v2.27.0)[69]. In case of multiple mapped reads, the number of multiple mapped reads was evenly distributed to the multiple mapped positions. For a read coverage plot, the SARS-CoV-2 genome was partitioned into 500 bins with each bin corresponds to ~60-nt region and the maximum coverage value for each bin was chosen (Figs. 2a and 3a and Supplementary Figs. 2a–d and 3a). Otherwise, it was plotted at a single nucleotide resolution (Supplementary Fig. 2f–h). For depicting PRF across ORFs 1a and 1b, we applied moving average with 4-kb sliding windows to smoothen the coverage plots (Supplementary Fig. 2i).

**Quantification of host and viral gene expression levels**. To quantify human gene expression using RPF-seq, QTI-seq, and mRNA-seq data, the number of reads per kilobase of exon per million mapped reads (RPKM) was calculated for each gene. Since all of the viral mRNAs start with 5′ leader sequence and contain identical sequences until the TRS-L at 70th nt position of the SARS-CoV-2 genome, the identity of a transcript can be determined by examining the sequence downstream of TRS-L. We collected mRNA-seq reads whose 5′ ends are mapped to the 5′ end of the SARS-CoV-2 genome, and calculated the number of reads per million mapped reads (RPM) of gRNA and each of sgRNAs (Fig. 2b). Since our mRNA-seq read lengths are long enough (100 nts) to be uniquely mapped, the numbers of the reads reflect the actual viral mRNA expression levels.

For ORF 10, TRS-B does not exist at upstream of the ORF and thus its canonical sgRNA is not produced by fusion between TRS-L and TRS-B. Hence, the reads where the junction occurs in the region between the 5′ end of ORF 10 and the 3′ end of upstream ORF N were counted to estimate ORF 10 sgRNA expression level.

RPF expression levels of SARS-CoV-2 genes were measured by obtaining RPF-seq and QTI-seq RPMs between the 5′ end of each annotated ORF and its 15-nt downstream position (Fig. 2b). Based on our analysis, RPF-seq reads mapped >15 nts downstream from the 5′ end of SARS-CoV-2 ORFs did not show the clear pattern of the expected 3-nt frame periodicity of mapped read positions[19] and therefore these reads were discarded when measuring RPF expression level. Thirteenth nt position of the RPF-seq or QTI-seq reads was used to determine the genomic position of the reads, which corresponds to the P-site of ribosome[70]. RPF expression level of ORF 1b was estimated by the expression level of ORF 1a multiplied by the PRF efficiency between ORFs 1a and 1b. RPF expression level for 3′UTR was also calculated by obtaining RPF-seq and QTI-seq RPMs between the 5′ end of 3′UTR and its 15-nt downstream position, and was used as a negative control to determine effective translation. TE for viral genes was calculated as RPF-seq RPM divided by mRNA-seq RPM, using 1 as a pseudocount. When calculating TE of ORF 7b, its RPF expression level was divided by mRNA expression level of ORF 7a sgRNA, rather than that of ORF 7b sgRNA, because translation of ORF 7b mainly takes place by leaky scanning of ribosome in ORF 7a sgRNA[45]. Canonical ORF 10 sgRNA is not clearly defined and detailed mechanisms of its translation have not been elucidated. Therefore, we calculated the TE of ORF 10 as the RPF expression level divided by the mRNA expression level of noncanonical ORF

10 sgRNA, and also by the mRNA expression level of N sgRNA, as the leaky scanning of ribosome in N sgRNA might lead to the translation of ORF 10.

**Detection of SARS-CoV-2 miRNA candidates and their transcriptome responses in human.** To detect miRNA candidates on the SARS-CoV-2 genome, we applied following criteria. First, we selected the windows predicted to be structured as hairpins. To select the potential hairpin forming region, RNA secondary structures were predicted to all overlapping 100-nt windows across the genome using RNAFold in ViennaRNA package with default parameters[71]. For each 100-nt window, hairpin-like score (H-score) was computed to measure the fraction of base pairs formed between the upstream and downstream 50-nt fragments of the window. For all $(i, j)$ pairs, we counted the number of pairs with $i < 50 \leq j$, where $i$ and $j$ are positions of paired nucleotides in upstream and downstream 50-nt fragments, respectively. The H-score ranges from 0.0 (unstructured) to 1.0 (structured as a perfect hairpin shape). We applied 0.5 to all windows as a minimum cutoff for the score (H-score ≥ 0.5). Second, the windows with high miRNA expression level (RPM > 1,000) were selected. To quantify the expression level, small RNA-seq sample at 36 hpi was used due to the highest number of mapped reads on the SARS-CoV-2 genome. Third, within each window that has met the above first and second criteria, the nt position to which the largest number of the 5′ ends of reads were mapped was chosen. The largest number of the 5′ ends of reads was divided by the total number of reads within the 100-nt window and this fraction was defined as homogeneity. If homogeneity is >0.5, we selected the nt position as the 5′ end of a mature miRNA candidate (Fig. 2c).

To monitor human transcriptome responses to the SARS-CoV-2 miRNA candidates, we utilized mRNA-seq datasets of uninfected and infected conditions at 36 hpi to match the sRNA-seq dataset used for miRNA detection (Fig. 2c). The human gene expression levels were measured as RPKM and the measured values from multiple replicates were averaged to reduce the noise. Top 50% of the most highly expressed mRNAs in the uninfected condition were used. Expression fold changes of each mRNA after viral infection were measured by comparing the expression levels between uninfected and infected conditions at 36 hpi. To observe the expression fold change by the identified SARS-CoV-2 miRNA candidates, we selected human mRNAs containing a single 7, 8mer target site in their 3′UTRs as targets, and defined mRNAs without any canonical site type (8mer, 7mer-m8, 7mer-A1, or 6mer) in their 3′UTRs as nontargets ("no site" group). Distribution of $\log_2$(mRNA fold change) of the target mRNAs was compared with that of nontargets by Wilcoxon's rank-sum test of "SciPy" package in Python (Fig. 2d).

**Detection of TIS-L.** To determine whether the RPF-seq and QTI-seq reads mapped to TIS-L at 59th nt position of the SARS-CoV-2 genome are derived from gRNA or sgRNAs, downstream sequences were required since all viral transcripts contain identical sequences until 75th nt position. However, since >90% of our RPF-seq and QTI-seq reads were <29 nts, most of TIS-L reads were multiple mapped to SARS-CoV-2 gRNA and sgRNAs. Therefore, we collected a subgroup of long reads that the P-site of ribosome (13th nt position) is mapped to TIS-L and its next codon (59th–64th nt positions) and that span to at least 3 nts downstream of TRS. Although the RPF-seq reads mapped to the codon next to TIS-L do not exactly have TIS-L in the P-site, these reads are highly likely to be RPFs of elongating ribosomes whose translation starts at TIS-L and clearly exhibit triplet nucleotide periodicity (Fig. 3b). Therefore, we decided to include these reads in this analysis. Most of the collected reads were uniquely mapped to the SARS-CoV-2 genome, and even multiple mapped reads were mapped <4 times. In case of these multiple mapped reads, the number of multiple mapped reads was evenly distributed to each of the mapped transcripts. By utilizing the number of long and uniquely reads, the relative abundance of each ORF was computed compared to the rest of ORFs and this relative abundance was extrapolated to all TIS-L-mapped reads, most of which are short and ambiguously mapped (Fig. 3c, d and Supplementary Figs. 3–6). The relative abundance of each ORF was consistently estimated when analyzing the reads exactly mapped to C nucleotide position of TIS-L (Supplementary Fig. 3e, f), and regardless of whether the host cells were partially contaminated by mycoplasma or not (Supplementary Fig. 5).

When analyzing RPF-seq reads of TIS-L with relaxed RNase I concentration (Fig. 3d, e), 14th nt position was used for the analysis as P-site of ribosome, which showed the best triplet periodicity and enrichment of reads at start codons of human genes. As these reads had longer lengths and blunt peaks near TISs due to the low RNase I concentration, we collected a subgroup of long reads that span to at least 3 nts downstream of TRS from a broader region (53rd–64th nt positions), and repeated the analysis described above.

To accurately measure the RPF expression level of ORF S by adding the translation from TIS-L (Fig. 3g and Supplementary Fig. 4e) at each time point, the number of RPF reads mapped between within TIS-L and its 15-nt downstream position was normalized by the relative abundance of ORF S computed from long reads. The relative abundance of ORF S at 48 hpi was used to calculate the RPF expression level of ORF S at 0–16 hpi, due to the small number of long reads (<30) mapped to TIS-L for these time points. The measured RPF expression level by TIS-L was added to the RPF expression level calculated from the annotated start position of ORF S. When visualizing the RPF expression level of each viral transcript (Fig. 4b–e and Supplementary Figs. 7 and 8), RPM at 5′ leader region upstream of TRS-L was also normalized by the relative abundances calculated from

long reads. The signal peptide of the S protein was predicted using SignalP-5.0[41] (Supplementary Fig. 7h).

**Plasmid construction.** To generate the plasmids used in the luciferase assay (Fig. 4f), the DNA fragments corresponding to 5′UTR sequences of subgenomic mRNAs for ORFs S, 6, and 7a (with or without CUG to CCG mutation) were in vitro synthesized (Integrated DNA Technologies) and inserted immediately upstream of RLuc gene of pRL-CMV (Promega). All the constructs were confirmed by DNA sequencing. All the primers used in this study are listed in Supplementary Table 2.

**DNA transfection and luciferase reporter assay.** Calu-3 cells were transiently transfected with various reporter plasmids described above using Lipofectamine 2000 (Life Technologies). Transient transfection was carried out according to the manufacturer's instructions. After 12 h, the cells were either uninfected or infected with SARS-CoV-2 at MOI 10 and incubated for 36 h. The incubated cells were harvested and resuspended in 200 μl of cold PBS. The resuspended samples were subjected to RNA extraction or luciferase assay, respectively. RLuc and FLuc activities were analyzed by using dual luciferase assay kit (Promega) and Glomax 20/20 Luminometer (Promega).

**RNA preparation and qRT-PCR.** Isolation of total-cell RNAs and complementary DNA (cDNA) synthesis was performed as described before[72–74]. After extraction of total RNAs using TRIzol Reagent (Thermo Fischer Scientific), the samples were digested with 0.05 U/μl DNase I (Thermo Fischer Scientific) for 45 min at 37 °C. Then, the DNase I-treated RNA samples were subjected to acid phenol/chloroform extraction. For cDNAs synthesis, the purified RNAs were mixed with 6 U/μl RevertAid reverse transcriptase (Thermo Fisher Scientific) and incubated at 37 °C for 2 h. qRT-PCR analysis was performed with the synthesized cDNAs, gene-specific oligonucleotides, and the LightCycler 480 SYBR Green I Master Mix (Roche) using LightCycler 480 II (Roche).

**Statistical test for luciferase assay results.** Two-tailed equal-variance Student's $t$ test was performed for statistical analysis with significance defined as a $P$ value < 0.05 (high significance at $P < 0.01$) (Fig. 4f and Supplementary Fig. 9e). Data obtained from three biological replicates were analyzed, unless indicated otherwise in figure legends, and were presented as the mean ± standard deviation. All raw data are provided in Source Data.

**Identification of DEGs.** The human gene expression levels measured as RPKM of RPF-seq were compared between before and after SARS-CoV-2 infection. By using DESeq2[75], expression fold change and the statistical significance as false discovery rate (FDR)-corrected $q$ value were calculated for each gene. A subset of genes with $q < 0.05$ were determined as upregulated and downregulated DEGs if $\log_2$(expression fold change) is >2.0 and <−2.0, respectively. The DEGs were identified for each hpi (Fig. 5a), and the DEGs detected at least one time point were collected. The DEGs with similar temporal expression patterns were grouped by hierarchical clustering, using "SciPy" package in Python (Fig. 5b, c and Supplementary Data 1). For mRNA-seq and QTI-seq data, a similar analysis was iterated (Supplementary Fig. 10a–c).

To investigate the temporal expression changes of the genes related to SARS-CoV-2 infection and host response, lists of genes were collected from previous studies (Fig. 6a–e). First, we gathered 332 genes that code proteins interacting with SARS-CoV-2 proteins[56] in addition to ACE2 and TMPRSS2. From a previous study[59], we obtained DEGs involved in type I IFN signaling or involved in cytokine activity. Genes with |$\log_2$(expression fold change)| > 1.0 in their mRNA-seq or genes with a significant difference of expression level in ELISA experiment were collected and their temporal expression changes were investigated on our mRNA-seq, RPF-seq, and QTI-seq data. Based on a previous study that identified host factors critical for SARS-CoV-2 infection in human lung cells[57], we collected host factors with $\log_2$(CRISPR screening fold change) > 1.0 at MOI 0.3 and measured temporal expression changes of these host factors. Availability of targeting drugs for each factor was curated from the Drug Gene Interaction database[76]. Across the time points, Spearman correlation coefficient was calculated between the rank of CRISPR screening for each host factor and the maximum value of RPF-seq |$\log_2$(expression fold changes)| with "SciPy" package in Python. The genes that affect the TI at CUG (eIF5, eIF5B, eIF2A, and eIF2D, eIF1, and eIF1A) were collected from a previous study[33] and their temporal expression patterns were investigated (Fig. 6i).

In order to identify differentially expressed miRNAs from our sRNA-seq data that have one replicate, we devised a statistical method as following: we counted the numbers of sRNA-seq reads mapped on annotated human pri-miRNAs and its surrounding regions with a flanking size of 1Mbp in both directions between uninfected and infected samples to build 2 × 2 contingency tables. $\chi^2$ test was performed for each of the constructed 2 × 2 contingency tables with "SciPy" package in Python. A subset of miRNAs with $q < 0.05$ were determined as upregulated and downregulated DEGs if $\log_2$(expression fold change) is >3.0 and <−3.0, respectively (Fig. 6h).

**GO enrichment and DAVID pathway analysis**. For each DEG cluster with similar expression patterns identified in the analysis of Fig. 5, biological processes enriched in the cluster were identified and visualized by using ClueGO[77,78]. Default parameters were used and up to five of the most significantly associated GO groups with $P < 0.05$ were displayed (Fig. 5d–g and Supplementary Figs. 10d and 11a–e).

GO enrichment analysis was also performed for DEGs at each time point (0–36 hpi) by using PANTHER with default parameters[79,80]. The most significantly associated three GO terms with FDR-corrected $q < 0.1$ at each time point were displayed (Fig. 6f and Supplementary Fig. 11f). Similarly, DAVID for KEGG pathway analysis[81] was performed for DEGs at each time point with default parameters. The most significant six KEGG pathways for DAVID analysis with FDR-corrected $q < 0.1$ at each time point were displayed (Fig. 6g and Supplementary Fig. 11g). Pathways likely irrelevant to viral infection such as bacterial infection pathways were discarded. During late phase (12–36 hpi), genes with $q < 0.01$ and $|\log_2(\text{expression fold change})| > 2.0$ were determined as DEGs. During early phase (0–4 hpi), since a small number of genes were marginally significant, genes with $q < 0.1$ and $|\log_2(\text{expression fold change})| > 1.5$ were determined as DEGs.

**Multiple sequence alignment of betacoronavirus genomes**. To obtain an evolutionary insight about translation from TIS-L, genome sequences of two representative betacoronaviruses from each of four lineages were collected from NCBI database[82,83] (Fig. 6j). The 5′ end sequences (140 nts) of viral genomes were aligned using ClustalW[84]. To further investigate whether TIS-L exclusively exists in the lineage B, genome sequences for more various betacoronaviruses were collected from NCBI database, and multiple sequence alignment was performed as above (Supplementary Fig. 11i).

**Reporting summary**. Further information on research design is available in the Nature Research Reporting Summary linked to this article.

## Data availability
The raw sequencing data, expression levels, and fold changes used in this study are available in the Gene Expression Omnibus database under accession number GSE157490. Source data used for creating all figures are provided as a Source Data file with this paper.

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

## Acknowledgements

This study was supported by the National Research Foundation of Korea (NRF) funded by the Ministry of Science and ICT, Republic of Korea (NRF-2014M3C9A3063541, NRF-2015R1A3A2033665, NRF-2017M3A9E4061995, NRF-2018R1A5A1024261, NRF-2018M3A9H4056537, NRF-2019M3E5D3073104, NRF-2020R1A2C3007032, and NRF-2020R1A5A1018081) and from Korea Health Industry Development Institute (KHIDI), funded by the Ministry of Health and Welfare, Republic of Korea (HI15C3224).

## Author contributions

Conceptualization and methodology: M.-S.P., Y.K.K., and D.B. Software and bioinformatic analysis: D.K., S.K. and J.A. Validation: J.P. and J.C. Investigation: J.P., H.R.C., J.C., H.P., J.U.P., N.S., G.K. and J.K. Data curation: S.K. Writing—original draft: H.R.C. and D.B. Writing—review and editing: D.K., S.K., J.P., H.R.C., J.C., J.A., H.P., M.-S.P., Y.K.K. and D.B. Visualization: D.K., S.K. and J.A. Supervision and project administration: K.K., M.-S.P., Y.K.K. and D.B. Resources and funding acquisition: M.-S.P., Y.K.K. and D.B.

## Competing interests

The authors declare no competing interests.
