## [Peer Review File · Nature Communications]

First round comments -

Reviewer #1 (Remarks to the Author):

The authors have constructed large datasets to understand the translome and transcriptome dynamics of SARS-CoV-2 using different established sequencing techniques (e.g. mRNA-seq & RPF-seq). The manuscript provides detailed insight into the temporal sequences of the activeness of transcription and translation in the Calu-3 model, which is commonly used in the study of SARS-CoV-2. Overall, the conclusions of the study are supported by the data; and the manuscript is well-written. However, the manuscript may be of interest only to a narrow scope of readers.

Although there are some interesting initial findings regarding the translome and transcriptome dynamics of SARS-CoV-2 (e.g. CoV-2 translation initiates at a previously unknown translation initiation site located in the leader sequence, TIS-L), the significance of the "altas" towards the understanding of the disease physiology is questionable - it maybe of limited interest to researchers in the field of SARS-CoV-2 biology in general. The authors state in the conclusion section that the "altas" may be useful for revealing the pathogenicity of the virus and therapeutic developments; however, it remains unclear that how this could be of use for practitioners in the field. The authors might consider demonstrating (at least) one application of their findings by using an illustrated example related to the actual pathophysiology of the disease.

Some specific comments:

1. As the authors pointed out, Vero cells are African green monkey kidney cells that are incapable of synthesizing IFNs and its lack of immune defense against viruses could be causing the inconsistencies between Calu-3 and Vero. So it is clear that Vero cell is not a good model for the current study. However, the authors still used Vero cell to "confirm whether this observed phenomenon is general, we infected the Vero cell line with SARS-CoV-2." The authors should use another physiologically-relevant model if they want to confirm the generality of their observation. The use of human lung organoid model could enhance the quality and impact of the current work.
2. The author speculated that the TIS-L creates an uORF that is overlapping and out-of-frame with the primary ORF of most other viral gene; they postulated this would mainly be a negative regulator of translation - how does this go with the postulation that it is an efficient TIS at the same time?
3. The human transcriptome/translatome part appears to be disconnected from the viral counterpart. Also it appears to be routine workflow and no significant novelty could be identified.

Reviewer #2 (Remarks to the Author):

Kim and colleagues describe a substantial analysis of SARS-CoV-2 gene expression during infection of tissue culture cells. The datasets are extensive, documenting the results of genome wide, high resolution probing of viral and cellular transcription and translation and the host response to infection. In principle, the manuscript is an advance on recently published deep sequencing analysis of the SARS-CoV-2 transcriptome (e.g. Kim et al., 2020; PMID 32330414) and translome (Finkel et al., 2020; PMID 32906143). However, although I admire the huge amount of work that has gone into this study, there are problems with the ribosome profiling datasets. Further, a recurrent issue throughout the manuscript is insufficient reporting of data quality. Without this, we cannot be sure that what is presented truly reflects the situation in infected cells. I have provided a number of examples of this below.

1. Quality control analysis of the ribosome-protected fragment (RPF) libraries is not provided. There are no plots of read numbers versus read lengths for the libraries so it is not clear if there is a defined peak corresponding to true RPFs, or whether there is contamination from untranslated viral mRNAs, which would show a broad size distribution corresponding to the size of the gel slice

taken during RPF purification. This kind of contamination, which can also arise from co-purifying cellular or viral RNPs, is especially a problem in viral ribosome profiling, where free viral RNA is in vast excess to translated RNA. Without these plots, we cannot be sure that the authors are simply mapping what are essentially RNA-Seq reads rather than RPF-Seq reads. The authors state that the average length of RPFs seen is 28 nt, but this doesn't confirm that there is a discrete peak (that one would get from true RPFs [at ~29/30 nt]).

2. The concern above arises (partly) from the fact that the authors show no evidence that the RPFs have reading frame periodicity. RPFs in most profiling datasets have a preference for the translated reading frame, with the majority of reads mapping to just one of the three possible frames. If there is no periodicity, there are generally two explanations. The first is that the cell lysates have been overtreated with RNase I leading to damage to the ribosomes and random trimming of the "protected" mRNA ends. I note that the authors state that the RPF length average is 28 nt, smaller than usual, and that the amount of RNase I added seems to be very high (5U/ μ l lysate). The second, more worrying explanation is that there is contamination from degraded (untranslated) viral RNA. I suspect that both of these factors may be present (see below).

3. My worry about viral RNA contamination is derived from examination of the RPFs mapping to ORFs 1a and 1b. Coronaviruses utilise programmed ribosomal frameshifting (PRF) to express the 1b ORF and as the efficiency of PRF is typically ~30-50%, there is reduced ribosome translation of the 1b ORF compared to 1a. However, as can be seen in Figure 2A, the read depth on ORF1a and ORF1b appears to be very similar, suggestive of ~100% frameshifting. This contrasts markedly with other published datasets (including the recent paper on SARS-CoV-2 profiling [Finkel et al. 2020]), where estimated PRFs are more typically 30-50%. This raises the serious concern that the authors are not mapping genuine RPFs, but RPFs contaminated with viral mRNA fragments, which would show similar read coverage on 1a and 1b. It is also apparent from other traces in this Figure that all is not as it should be. The read coverage seen for RPF-Seq, QTI-Seq and sRNA-Seq are surprisingly similar. This is odd, especially for QTI-Seq, where one should see clearly defined peaks of translation initiation at the start codon of the 1a ORF, and at the start codons of the structural and accessory genes. This is not seen here, but has been seen in other coronavirus profiling studies where initiation has been specifically examined using harringtonine (Irigoyen et al. [Ref. 44]).

4. The mRNA-Seq data is barely discussed. Surely there must be some insights into, for example, utilisation of standard and non-standard body transcription regulatory sequences? Additionally, there do not appear to be many reads corresponding to the 5' leader sequence, which should be the most abundant peak in both positive and negative strand RNA plots?

5. The authors note mycoplasma contamination (Extended Data fig 1). This could be seen as a serious problem for the study, but we are not told what the proportion of contaminating reads is. A table of read numbers for the various classes of RNA would be informative.

6. A main conclusion of the study is the translational utilisation of a CUG codon in the viral leader region at the 5' end of the genome. The authors note briefly that this has been seen in another study (ref 44), but it has also been documented for the coronavirus IBV (Dinan et al., 2019; PMID 31243124) and in SARS-CoV-2 itself (Finkel et al., 2020). I am very surprised that the latter manuscript wasn't mentioned at all in this submission!

Reviewer #3 (Remarks to the Author):

The manuscript "A high-resolution temporal atlas of the SARS-CoV-2 translate and transcriptome" by Kim and colleagues provides a thorough and comprehensive analysis of viral and host transcription, and translation in Calu-3 cells during the life cycle of SARS-CoV-2 replication. The amount of data is overwhelming and of very good quality. The authors analysed several early (0, 1, 2, 3, 4 hours post infection) and late (12, 16, 24, 36 hours post infection) for mRNAseq, RPF-seq, QTI-seq and sRNA-seq. The obtained data provide a very detailed overview on virus replication and host responses. The authors chose to depict individual observations, which make it

difficult for the reader to see the overall picture. The data however are compelling and the authors may like to address the following comments.

1. As mentioned above, the manuscript would gain more clarity if some more general findings and observations are displayed. Although there're already ribosomal profiling data published, the current data set seems to be of higher quality and the authors should consider to display an overview on all SARS-CoV-2 ORFs that are translated. Although this may look very similar to published work, it would, first, illustrate the quality of the data and, second, allow the reader to get into the topic by providing a global overview on the translome.

2. After providing a more general overview, the reader would be prepared to "digest" the more specific findings, such as the leader TIS. This part on the TIS-L is taking a prominent part of the entire manuscript, however, the overall biological significance is not clear (see below). Therefore, an introduction with a more general overview would raise the acceptance of the reader to get into the more speculative TIS-L part.

3. While the TIS-L may have some role to play in SARS-CoV-2 replication, it is not clear if this mechanism is relevant (for the scope of this manuscript it is acceptable to describe the observation) for SARS-CoV-2 or other CoVs. Some statements should be adjusted since there's no evidence about the impact of the TIS-L. For example, the statement that the "TIS-L has a substantial regulatory impact on most SARS-CoV-2 ORFs..." is not supported by the provided data. It should be phrased in a more speculative way. It is not clear if a 2.6- or 2-fold promotion or reduction of a reporter protein expression is meaningful for viral gene expression.

4. An overview on the position of TIS-L in relation to start codons of viral ORFs on viral genomic and sg mRNAs should be provided as an overview figure panel.

5. Since expression of ORF10 is still under debate in the field, the authors may summarise some data regarding the detection of ORF10-containing mRNA (does it exist or not). Apparently, ORF10 is translated but there's still the question if a separate sg mRNA exists. The authors may contribute to this discussion with their data and some words in the discussion.

6. The individual figures are very data rich and some details are only rudimentary (or not) mentioned in the text. It would be good to adjust the main text to accommodate all these interesting findings.

7. A more elaborate discussion would help the reader to judge on the impact of individual findings.

Response to reviewers' comments

We would like to thank the reviewers for the careful and thorough evaluation of this manuscript and for the insightful comments and constructive suggestions, which helped us to improve the quality of the manuscript. Please see below, in blue, our detailed response to reviewers' comments.

Reviewer #1

(R1-1) The authors have constructed large datasets to understand the translome and transcriptome dynamics of SARS-CoV-2 using different established sequencing techniques (e.g. mRNA-seq & RPF-seq). The manuscript provides detailed insight into the temporal sequences of the activeness of transcription and translation in the Calu-3 model, which is commonly used in the study of SARS-CoV-2. Overall, the conclusions of the study are supported by the data; and the manuscript is well-written. However, the manuscript may be of interest only to a narrow scope of readers.

Although there are some interesting initial findings regarding the translome and transcriptome dynamics of SARS-CoV-2 (e.g. CoV-2 translation initiates at a previously unknown translation initiation site located in the leader sequence, TIS-L), the significance of the "altas" towards the understanding of the disease physiology is questionable - it maybe of limited interest to researchers in the field of SARS-CoV-2 biology in general. The authors state in the conclusion section that the "altas" may be useful for revealing the pathogenicity of the virus and therapeutic developments; however, it remains unclear that how this could be of use for practitioners in the field. The authors might consider demonstrating (at least) one application of their findings by using an illustrated example related to the actual pathophysiology of the disease.

We would like to thank the reviewer for the valuable advice. Following the reviewer's suggestion, we revised our manuscript by portraying several examples of how our SARS-CoV-2 translome and transcriptome dataset and how the findings from our study can be utilized to understand the pathophysiology of the SARS-CoV-2 and to develop therapeutic strategies as follows.

First, our temporal atlas can be a valuable resource in the investigation for potentially effective therapeutic targets, as our translome dataset provides precise measurements of the gene expression response of the host upon SARS-CoV-2 infection. Since the translome contains the integrated abundance of transcripts and their translational regulation, measurement of gene expression levels using the translome dataset is considered to better reflect the actual expression level of proteins in cells, compared to the transcriptome dataset. For instance, our translome dataset illustrates that the previously reported decrease in ACE2 expression upon viral infection (Tay *et al.*, 2020, Bojkova D *et al.*, 2020, and Kuba *et al.*, 2006) was mediated primarily by translational regulation rather than transcriptional regulation (**Fig. R1-1a**), demonstrating that our translome dataset helps reveal more detailed mechanism of gene expression of a key protein, ACE2. Utilizing this accurately measured gene expression response of the host upon SARS-CoV-2 infection, we investigated the temporal expression patterns of host factors identified to be critical for SARS-CoV-2 infection (Daniloski *et al.*, 2021) (**Fig. R1-1a**). As a result, the differential expression upon viral infection for multiple host factors, including the targets of existing drugs was identified (magenta, **Fig. R1-1a**). Intriguingly, we discovered the magnitude of differential expression of the host factors was significantly associated with the impact of each factor on viral infection (**Fig. R1-1b**). This result indicates that the host factors with larger changes in their expressions upon SARS-CoV-2 infection tend to be more critical for SARS-CoV-2 infectivity, demonstrating how our dataset can be utilized in the selection of more effective therapeutic targets.

Our dataset can also be useful for a deeper understanding of the molecular mechanism of SARS-CoV-2 pathophysiology. In our study, we report that translation of SARS-CoV-2 viral ORFs is globally regulated by TIS-L, a non-canonical CUG start codon. Since several eukaryotic translation initiation (TI) factors have been identified to affect the rate of the TI at the non-canonical start codons (Kearse and Wilusz, 2017) and the viral translation relies on the host machinery, we have examined the temporal expression patterns of these TI factors upon SARS-CoV-2 infection. Accordingly, we discovered that eIF5 unlike other TI factors was particularly upregulated in the late phase after viral infection (**Fig. R1-1c**). This result is also consistent with the previous reports that the increase of eIF5 level promotes non-canonical TI (Loughran *et al.*, 2012; Barth-Baus *et al.*, 2013). We postulate that eIF5 may be a key TI factor for the active TI at TIS-L, potentially leading to a more severe pathophysiological consequence of SARS-CoV-2 infection compared to other betacoronaviruses. This illustrated example demonstrates our temporal atlas can be utilized to identify important TI factor candidates responsible for SARS-CoV-2 infection and thus delineate the more detailed molecular mechanisms of SARS-CoV-2 pathophysiology. Moreover, TIS-L-mediated regulation of the SARS-CoV-2 translome may be applied to develop a novel therapeutic strategy. Since TIS-L appears to act as a master regulator of the SARS-CoV-2 translome, blocking this region with an antisense oligonucleotide (ASO) might lead to efficient therapeutics by disrupting the TIS-L-mediated regulation of the SARS-CoV-2 translome (**Fig. R1-1d**).

Taken together, these findings illuminate that our temporal atlas of the SARS-CoV-2 translome will be a valuable resource to develop novel therapeutics and to understand the pathophysiology of SARS-CoV-2. Thanks to the reviewer, we were able to greatly improve our manuscript by including these results.

Fig. R1-1. Utilization of the translome dataset for understanding the SARS-CoV-2 pathophysiology and developing therapeutic strategies.

a, Temporal expression changes of the host factors required for SARS-CoV-2 infection (brown), identified by CRISPR screening (Daniloski *et al.*, 2021). Host factors whose targeting drugs were found in the Drug Gene Interaction database (DGIdb) are highlighted in magenta. The grey lines represent the expression changes of individual DEGs identified in **Fig. 5**. For mRNA-seq (left), RPF-seq (middle), and QTI-seq

(right) data, $\log_2(\text{expression fold changes})$ for the genes at each time point were measured.

b, Association between the magnitude of differential expression and the impact on SARS-CoV-2 infectivity for the host factors investigated in **(a)**. The x-axis represents the maximum of the absolute values of RPF-seq $\log_2(\text{expression fold changes})$ across the time points. The y-axis represents the ranks of the CRISPR screening enrichment for the host factors (Daniloski *et al.*, 2021). Otherwise as in **(a)**.

c, Temporal expression changes of the genes reported to enhance (red) or suppress (blue) the translation initiation at non-AUG codons (Kearse and Wilusz, 2017). Otherwise as in **(a)**.

d, A hypothetical therapeutic strategy for SARS-CoV-2 treatment by blocking TIS-L with antisense oligonucleotide, which might disrupt the viral translation and reduce the viral infectivity.

Some specific comments:

(R1-2) As the authors pointed out, Vero cells are African green monkey kidney cells that are incapable of synthesizing IFNs and its lack of immune defense against viruses could be causing the inconsistencies between Calu-3 and Vero. So it is clear that Vero cell is not a good model for the current study. However, the authors still used Vero cell to "confirm whether this observed phenomenon is general, we infected the Vero cell line with SARS-CoV-2." The authors should use another physiologically-relevant model if they want to confirm the generality of their observation. The use of human lung organoid model could enhance the quality and impact of the current work.

To address the reviewer's concern, we have additionally performed RPF-seq and QTI-seq with Caco-2 cell line, which is derived from the human colon and is therefore physiologically more relevant than Vero. To examine whether the findings from our study are general enough to be applied to other cell types, we compared the expression levels of SARS-CoV-2 transcriptome between Calu-3 and Caco-2 cell lines. Accordingly, we found the overall expression patterns of viral ORFs were fairly similar to each other although the overall expression levels were slightly lower in Caco-2 (**Fig. R1-2a**). Distributions of RPF-seq and QTI-seq reads across the SARS-CoV-2 genome were also similar between the two cell lines (**Fig. R1-2b**). When focusing on the 5' leader region, consistent with the Calu-3 result, we observed that a considerable amount of RPF-seq and QTI-seq reads were mapped to TIS-L (**Fig. R1-2c**) in Caco-2. The estimated relative fraction of the TIS-L reads derived from each viral mRNA was also similar to that estimated in the Calu-3 dataset (**Fig. R1-2d**). Taken together, the overall consistency of the viral ORF expression along with the observed enrichment of RPF-seq reads at TIS-L in another human cell line clearly demonstrates that our findings in this study should be general enough to be applied to other cell types. Thanks to the reviewer, we have included these results on the generality of this work in the manuscript (**Fig. 3h and Supplementary Fig. 5a**).

Fig. R1-2. Comparison of the SARS-CoV-2 transcriptome between Calu-3 and Caco-2 cells.

a, Comparison of the RPF-seq (left) and QTI-seq (right) expression levels of SARS-CoV-2 ORFs between Calu-3 and Caco-2 cell lines at 48 hours post-infection (hpi). Expression levels of each ORF were measured as RPM and displayed as $\log_{10}(\text{RPM}+1)$.

b, Coverage of RPF-seq and QTI-seq reads across the SARS-CoV-2 genome for Calu-3 (top) and Caco-2 (bottom) cell lines at 48 hpi. The y-axis represents the number of reads per million mapped reads (RPM) on a \log_{10} scale. Blue and orange bars indicate the number of reads mapped to positive and negative strands of the genome, respectively.

c, Enrichment of RPF-seq (left) and QTI-seq (right) reads at the TIS-L for Calu-3 (top) and Caco-2 (bottom) cell lines at 48 hpi. The 13th position (12-nt offset from the 5' end) of the reads, indicating the ribosome P-site position, was counted and normalized to be the number of reads per million mapped reads (RPM). Open reading frames are depicted as three different colored bars with the dark blue bars indicating in-frame with TIS-L and the others out-of-frames.

d, Comparison of the relative fractions of RPF-seq (top) and QTI-seq (bottom) TIS-L reads uniquely mapped to each of gRNA and sgRNAs between Calu-3 and Caco-2 cell lines at 48 hpi.

(R1-3) The author speculated that the TIS-L creates an uORF that is overlapping and out-of-frame with the primary ORF of most other viral gene; they postulated this would mainly be a negative regulator of translation – how does this go with the postulation that it is an efficient TIS at the same time?

Given the reviewer's comment, we noticed that our explanation of how the TIS-L can function as a negative regulator of translation was not clear enough. As shown in **Figure 3b**, TIS-L is considered to be an efficient translation initiation site (TIS) as a large amount of RPF-seq reads were mapped to it. For sgRNAs of ORFs 3a, E, M, 7a, 7b, and 8, the uORF created by TIS-L is overlapping and out-of-frame with the primary ORF, which would make TIS-L function as a negative regulator for these ORFs. Since the TIS-L is an efficient TIS, a considerable number of ribosomes would recognize it and start the translation of a uORF. These ribosomes would stop at a stop codon of the uORF and cannot be used to translate the primary ORF, resulting in the reduction of translation initiation at primary ORF and a negative regulatory impact on its translation. Our experimental validation result supports our claim that TIS-L functions as a translational suppressor for ORF 7a (**Fig. 4f**). The translational regulatory effect by TIS-L is expected to be significant, based on our observation that the level of translation initiation of TIS-L was even higher than that of the annotated TISs for several ORFs from our RPF-seq and QTI-seq dataset (**Fig. 3f**). Collectively, our RPF-seq, QTI-seq, and luciferase assay results indicate TIS-L is an efficient TIS, and also a negative regulator of translation for ORFs overlapping and out-of-frame with the uORFs from TIS-L. Thanks to the reviewer, we have improved the description of TIS-L in our manuscript to reduce potential confusion.

(R1-4) The human transcriptome/translatome part appears to be disconnected from the viral counterpart. Also it appears to be routine workflow and no significant novelty could be identified.

To address the reviewer's concern, we have performed following analyses.

First, we have examined the temporal expression patterns of several translation initiation (TI) factors affected the rate of the TI at the non-canonical start codons, in order to investigate the connection between these factors and the active TI at TIS-L. Accordingly, as described in response to the comment **R1-1**, eIF5 was particularly upregulated in the late phase after viral infection (**Fig. R1-1c**). We postulate that eIF5 may be a key TI factor for the active TI at TIS-L, because the previous studies have reported the association between the increase in eIF5 level and promotion of non-canonical TI (Loughran *et al.*, 2012; Barth-Baus *et al.*, 2013).

Second, we have investigated the response of human transcriptome to the potential SARS-CoV-2 miRNAs identified in our sRNA-seq analysis (**Fig. R1-4a**). To monitor the human mRNA repression by the SARS-CoV-2 miRNA candidates, human mRNAs containing a single 7, 8mer target site of the identified candidates in their 3'UTRs were selected (**Methods**). Accordingly, significant repression was observed

upon viral infection from two candidates (**Fig. R1-4b**). Moreover, candidate A exhibited miRNA-like stem-loop structures and primary sequence determinants known to be required for proper processing of mature miRNAs, such as basal UG, apical UGUG, or downstream CNNC. This result raises a compelling possibility of an interplay between the viral miRNAs and human gene expressions. These analyses strengthen the connection between the virus and human at the transcriptome and translome levels with an interesting novel finding of a potential factor that may be associated with the non-canonical TI of viral ORFs, and a potential host gene regulatory network by viral miRNAs. Thanks to the reviewer, we were able to largely improve the manuscript and have included these results in our manuscript (**Fig. 6i and Fig. 2c, d**).

Fig. R1-4. SARS-CoV-2 miRNA candidates and their transcriptome responses in human

a, Two examples of potential miRNA candidates detected on the SARS-CoV-2 genome. sRNA-seq reads mapped on the SARS-CoV-2 genome are displayed as blue bars and the corresponding H-scores, that summarize the folding degree of the RNA hairpin structure centered on the nucleotide position (**Methods**), are depicted as a red line below. Predicted RNA secondary structures of the two miRNA candidates are also illustrated where predicted mature miRNAs are shown in blue and known determinants for miRNA processing are indicated.

b, Repression of human mRNAs targeted by SARS-CoV-2 miRNA candidates identified in (a). Expression fold changes of each mRNA after viral infection were measured by mRNA-seq for Calu-3 cells at 36 hpi. Human mRNAs containing a single 7, 8mer target site of the identified candidates in their 3'UTRs were selected (**Methods**). Cumulative distribution of $\log_2(\text{mRNA fold change})$ of the target mRNAs was plotted (red) and compared with that of non-targets ('No site', black) by Wilcoxon's rank-sum test.

Reviewer #2

Kim and colleagues describe a substantial analysis of SARS-CoV-2 gene expression during infection of tissue culture cells. The datasets are extensive, documenting the results of genome wide, high resolution probing of viral and cellular transcription and translation and the host response to infection. In principle, the manuscript is an advance on recently published deep sequencing analysis of the SARS-CoV-2 transcriptome (e.g. Kim *et al.*, 2020; PMID 32330414) and translome (Finkel *et al.*, 2021; PMID 32906143). However, although I admire the huge amount of work that has gone into this study, there are problems with the ribosome profiling datasets. Further, a recurrent issue throughout the manuscript is insufficient reporting of data quality. Without this, we cannot be sure that what is presented truly reflects the situation in infected cells. I have provided a number of examples of this below.

(R2-1) Quality control analysis of the ribosome-protected fragment (RPF) libraries is not provided. There are no plots of read numbers versus read lengths for the libraries so it is not clear if there is a defined peak corresponding to true RPFs, or whether there is contamination from untranslated viral mRNAs, which would show a broad size distribution corresponding to the size of the gel slice taken during RPF purification. This kind of contamination, which can also arise from co-purifying cellular or viral RNPs, is especially a problem in viral ribosome profiling, where free viral RNA is in vast excess to translated RNA. Without these plots, we cannot be sure that the authors are simply mapping what are essentially RNA-Seq reads rather than RPF-Seq reads. The authors state that the average length of RPFs seen is 28 nt, but this doesn't confirm that there is a discrete peak (that one would get from true RPFs [at ~29/30 nt]).

(R2-2) The concern above arises (partly) from the fact that the authors show no evidence that the RPFs have reading frame periodicity. RPFs in most profiling datasets have a preference for the translated reading frame, with the majority of reads mapping to just one of the three possible frames. If there is no periodicity, there are generally two explanations. The first is that the cell lysates have been over-treated with RNase I leading to damage to the ribosomes and random trimming of the "protected" mRNA ends. I note that the authors state that the RPF length average is 28 nt, smaller than usual, and that the amount of RNase I added seems to be very high (5U/ μ l lysate). The second, more worrying explanation is that there is contamination from degraded (untranslated) viral RNA. I suspect that both of these factors may be present (see below).

We would like to thank the reviewer for pointing out the importance of evaluating the quality of the RPF-seq and QTI-seq data. To address the reviewer's concern on possible contamination with untranslated viral RNA fragments, we have examined our translome data in terms of their length distributions and triplet periodicities.

First, we examined the length distribution of the libraries from our RPF-seq and QTI-seq data. Regardless of the time point, sequencing method, or mapped genome, we observed a discrete peak at 28~29 nt (**Fig. R2-1a**), that is the expected size of ribosome footprints. We have also confirmed that the distribution of the reads mapped to the SARS-CoV-2 genome was similar to that of the host genome, supporting that our RPF-seq and QTI-seq reads mapped to the SARS-CoV-2 genome are not contaminated by untranslated viral RNA fragments with variable lengths.

Second, we investigated whether the triplet periodicity of the mapped position of reads is observed for RPF-seq and QTI-seq data, which is one of the major characteristics of ribosome profiling. For all the experiments that we have performed, >50% of the reads were mapped to the first position of triplet codon (**Fig. R2-1b**). This clear periodicity also confirms the enrichment of footprints of ribosomes, neither

damaged by treatment of RNase I with high concentration nor contaminated by untranslated viral RNA fragments. The reviewer's suggestion helped greatly in proving the high quality of our dataset, and we have updated our manuscript to include the results of these quality check analyses.

Fig. R2-1. Quality analysis of RPF-seq and QTI-seq datasets

a, Distribution of read lengths for translatoome dataset. For RPF-seq (top) and QTI-seq (bottom) reads mapped to the human (green) or SARS-CoV-2 genome (red), distributions of the read lengths are shown for each time point and each cell line.

b, Triplet nucleotide periodicity of translatoome dataset. The 13th nucleotide position (12-nt offset from the 5' end) of the reads mapped to the human (green) or SARS-CoV-2 (red), indicating the ribosome P-site position, was counted for RPF-seq (top) and QTI-seq (bottom). The relative fractions of the ribosome P-sites mapped to each of three codon nucleotides are shown for each time point and each cell line. Open reading frames are depicted as three different colored bars with the darkest bars indicating in-frame and the others out-of-frames.

(R2-3) My worry about viral RNA contamination is derived from examination of the RPFs mapping to ORFs 1a and 1b. Coronaviruses utilise programmed ribosomal frameshifting (PRF) to express the 1b ORF and as the efficiency of PRF is typically ~30-50%, there is reduced ribosome translation of the 1b ORF compared to 1a. However, as can be seen in **Figure 2A**, the read depth on ORF1a and ORF1b appears to be very similar, suggestive of ~100% frameshifting. This contrasts markedly with other published datasets (including the recent paper on SARS-CoV-2 profiling [Finkel *et al.* 2021]), where estimated PRFs are more typically 30-50%. This raises the serious concern that the authors are not mapping genuine RPFs, but RPFs contaminated with viral mRNA fragments, which would show similar read coverage on 1a and 1b. It is also apparent from other traces in this Figure that all is not as it should be. The read coverage seen for RPF-Seq, QTI-Seq and sRNA-Seq are surprisingly similar. This is odd, especially for QTI-Seq, where one should see clearly defined peaks of translation initiation at the start codon of the 1a ORF, and at the start codons of the structural and accessory genes. This is not seen here, but has been seen in other coronavirus profiling studies where initiation has been specifically examined using harringtonine (Irigoyen *et al.* [Ref. 44]).

To address the reviewer's concern regarding the possibility of contamination by viral RNA, we have closely examined our translome data and evaluated its quality based on programmed frameshifting (PRF) and translation initiation sites (TISs). PRF is an important characteristic of coronavirus which should be detected in RPF-seq dataset. We realized that ~50% reduction of the read density can be attributable to the PRF at ORF1b but was not clearly visible in **Figure 2a**, because the read coverage was presented on a \log_{10} scale. When we compared the triplet periodicity of the mapped position of reads between ORFs 1a and 1b, +1nt shift of the reading frame was observed for ORF 1b (**Fig. R2-3a**). We also measured the PRF efficiency by comparing the read densities between ORF1a and ORF1b and obtained 64% efficiency on average (**Fig. R2-3b**), comparable to $57 \pm 12\%$ reported in a previous study (Finkel *et al.*, 2021). These results indicate that our RPF-seq dataset captured the true footprint of ribosomes and therefore the concern about viral RNA contamination can be alleviated.

The reviewer has raised another concern that the overall read coverages between RPF-seq and QTI-seq are similar, and that QTI-seq reads seem to be not enriched at TISs. Similar to the PRF, the enrichment of reads at TISs of ORFs was not clearly visible due to the \log_{10} scale in **Figure 2a**. To examine if our QTI-seq reads are enriched with reads from TISs, we quantitatively assessed the relative distribution of the reads within human and viral ORFs for RPF-seq and QTI-seq dataset. Accordingly, the QTI-seq reads were more enriched in the annotated TISs for ORFs compared to RPF-seq reads (**Fig. R2-3c**), which confirms the overall quality of our QTI-seq data. In addition, to more accurately observe the translation initiation, we reperfomed QTI-seq in the Calu-3 cell line with the following modifications. The incubation time for pre-treatment of 2 μ g/mL harringtonine was increased from 10 minutes to 20 minutes to more completely remove the pre-engaged ribosomes from mRNA. Then, the cells were lysed with 600 μ L of lysis buffer which contains 2 μ g/mL of harringtonine. With the updated method, enhanced read enrichment at TISs of ORFs from QTI-seq was observed (**Fig. R2-3d**). These results demonstrate that our original QTI-seq and improved QTI-seq both confirm reliable quality of our QTI-seq datasets. Thanks to the reviewer, we were able to prove the high quality of our translome dataset, and we included these results shown in **Fig. R2-3b, c** to the manuscript (**Fig. 1e and Supplementary Fig. 2i**).

Fig. R2-3. General features of SARS-CoV-2 translome dataset.

a, Triplet nucleotide periodicity of ORFs 1a and 1b. The 13th nucleotide position (12-nt offset from the 5' end) of the reads mapped to ORFs 1a or 1b, indicating the ribosome P-site position, was counted for RPF-seq dataset at 48 hours-post infection. The relative fractions of the ribosome P-sites mapped to each of three codon nucleotides are shown for each time point and each cell line. Open reading frames are

depicted as three different colored bars with the darkest bars indicating in-frame and the others out-of-frames to ORF 1a.

b, Efficiency of programmed ribosomal frameshifting (PRF) between ORF1a and ORF1b measured for each time point and each cell line.

c, Distribution of mRNA-seq (left), RPF-seq (middle), and QTI-seq (right) reads with respect to the relative position near the start of ORF. For 4 hpi and 36 hpi, the 13th nucleotide position (12-nt offset from the 5' end) of the reads mapped to the human (green) or SARS-CoV-2 (red) was counted for each sequencing data. The relative fraction to the amount of reads mapped to entire ORF was calculated for each position, and the y-axis represents the average of the relative fractions for ORFs with >50 reads mapped. Open reading frames are depicted as three different colored bars with the darkest bars indicating in-frame and the others out-of-frames.

d, Enrichment of QTI-seq reads mapped to the translation initiation sites compared to RPF-seq reads. For RPF-seq and QTI-seq data at 48 hpi with different cell lines and updated experiment protocol, relative fraction of the reads whose 13th nucleotide position was mapped to the first codon of ORFs was calculated and compared.

(R2-4) The mRNA-Seq data is barely discussed. Surely there must be some insights into, for example, utilisation of standard and non-standard body transcription regulatory sequences? Additionally, there do not appear to be many reads corresponding to the 5' leader sequence, which should be the most abundant peak in both positive and negative strand RNA plots?

We agree with the reviewer that our mRNA-seq data can provide more insights to understand the SARS-CoV-2 transcriptome and therefore we have added following analyses.

First, to quantify the amount of standard and non-standard body transcription, we collected the junctions detected in the discontinuous transcription of the SARS-CoV-2 genome. All detected junctions were divided into two groups by whether they were dependent on the previously annotated transcriptional regulatory sequence (TRS, canonical) or independent of TRS (non-canonical). Accordingly, 78.5% of the junctions were mapped to the canonical junctions which generate sgRNAs (**Fig. R2-4a**). This result is comparable to that of the previous report (Kim *et al.*, 2020).

Second, we investigated whether ORF 10 sgRNA is produced by non-canonical junctions, in order to help evaluate whether ORF 10 is expressed and functional. When measuring the number of junctions that occurred between TRS-L and the 5' end of ORF 10, a very small amount of the reads (<8 RPM), which corresponds to <0.02% of the total junction-spanning reads, were obtained (**Fig. R2-4b, top**). Moreover, the majority of these ORF 10 non-canonical junctions occurred downstream of ORF 10, and these hypothetical ORFs were out-of-frame with ORF 10 (**Fig. R2-4b, bottom**). From these observations, we conclude that the TRS-dependent junctions specific for ORF10 sgRNA are not produced during viral discontinuous transcription. However, translation initiation at the annotated TIS and an increased level of translation at 24 and 36 hpi was clearly detected for ORF 10 both from RPF-seq and QTI-seq results (**Supplementary Fig. 8b,c**), indicating that ORF 10 might produce a functional protein. These results collectively suggest the translation of ORF 10 mainly occurs from sgRNAs of the other ORFs, rather than from its own sgRNAs, perhaps by ribosomal leaky scanning or reinitiation after the termination of translation for upstream ORFs.

The reviewer also raised a concern that the enrichment of mRNA-seq reads at 5' leader region seems to be not observed from our dataset. This is again because log₁₀-scale was used for **Figure 2a**. When the

same plot was presented on a natural scale, the enrichment was obviously visible (**Fig. R2-4c**). As the reviewer expected, for the reads mapped to the positive strand, a clear peak was observed upstream of TRS-L. On the other hand, enrichment of reads at 5' leader region was not observed for the reads mapped to negative strand, due to their small amount of 0.1% of total mapped reads. Thanks to the reviewer for the suggestion, we were able to improve our manuscript by adding these analyses regarding the SARS-CoV-2 transcriptome (**Supplementary Figs. 2e,f and 8a**).

Fig. R2-4. Characteristics of SARS-CoV-2 transcriptome.

a, Canonical and non-canonical junctions observed in discontinuous transcription. Junctions mediated by transcriptional regulatory sequences (TRS-L and TRS-B) are annotated as 'canonical' and the others as 'non-canonical'. The number of reads per million mapped reads (RPM) was calculated for each junction, and relative fractions of junctions detected in all mRNA-seq samples at 36 hours post-infection (hpi) were presented.

b, Non-canonical ORF 10 sgRNAs. From mRNA-seq at 36 hpi, the reads that include a junction located in the region between 100 nt upstream and 50 nt downstream of ORF 10 start position were counted and calculated as RPM. The most abundant five junction pairs are shown with their genomic positions. ORF 10 sgRNAs created from each junction and their reading frames to ORF 10 are depicted below. Open reading frames are depicted as three different colored bars with the darkest bars indicating in-frame and the others out-of-frames with respect to ORF 10.

c, Read coverage plots of mRNA-seq for 5' end (position 0 – 300) of the SARS-CoV-2 genome at 36 hpi. The read depths of positive and negative strands were presented in blue and orange, respectively, on a natural scale of RPM. Translation initiation site located in 5' leader (TIS-L) and TRS-L are depicted as orange and yellow boxes, respectively.

(R2-5) The authors note mycoplasma contamination (**Extended Data Fig. 1**). This could be seen as a serious problem for the study, but we are not told what the proportion of contaminating reads is. A table of read numbers for the various classes of RNA would be informative.

To address the reviewer's concern, a detailed table with the number of discarded reads at each pre-processing step of our NGS pipeline is presented below (**Table R1**). The number of reads mapped to non-coding transcripts (e.g., rRNA and tRNA) and the mycoplasma genome are provided in this table. Overall, ~10% of sequenced reads in mRNA-seq samples and ~3% in RPF-seq, QTI-seq, and sRNA-seq samples were mycoplasma-derived reads in the Calu-3 cell line. For our analysis, we have filtered out these reads and have used the remaining reads (**Methods**).

Furthermore, in order to alleviate the reviewer's concern that mycoplasma contamination may have disturbed our result, we have cured the Calu-3 cell line from mycoplasma contamination and performed RPF-seq and QTI-seq. The treatment seems to be successful as only a negligible amount of reads were mapped to the mycoplasma genome (**Table R2**). When compared the expression levels of the SARS-CoV-2 and human transcriptomes between the previous and mycoplasma-cured data, we found the overall expression patterns of ORFs were fairly similar to each other (**Fig. R2-5a,b**). For the transcriptome dataset from the cured cells, >10% of the SARS-CoV-2 reads were mapped to TIS-L, and the relative fractions of TIS-L reads uniquely mapped to each of gRNA and sgRNAs were also comparable to the results from the previous data (**Fig. R2-5c,d**). Following the reviewer's suggestion, we have included the supplementary tables with detailed numbers of the discarded reads (**Supplementary Table 1**).

assay	infection	replicates	sequenced	step1 (qualtrim)	step2 (adaptrim)	step3 (artifact)	step4 (ncRNA)		step5 (myco)	
mRNA-seq	U	10	288,542,388	10,482,052	2,566,896	6,032	122,286,109	42.4%	49,169,090	17.0%
mRNA-seq	0	4	109,615,974	3,670,486	490,425	3,074	39,238,160	35.8%	18,962,281	17.3%
mRNA-seq	1	4	119,225,613	3,745,977	636,468	2,797	46,524,335	39.0%	19,683,560	16.5%
mRNA-seq	2	4	107,587,508	3,381,038	612,091	2,803	44,741,859	41.6%	16,425,309	15.3%
mRNA-seq	4	4	104,210,824	3,217,492	1,413,919	2,703	49,416,613	47.4%	12,569,073	12.1%
mRNA-seq	12	4	80,172,520	2,667,146	3,790,112	1,286	33,589,630	41.9%	11,096,699	13.8%
mRNA-seq	16	4	81,862,979	2,903,917	5,263,966	1,137	40,729,328	49.8%	11,155,841	13.6%
mRNA-seq	24	4	76,850,194	2,481,836	2,331,072	847	35,896,365	46.7%	12,553,905	16.3%
mRNA-seq	36	4	77,368,984	2,477,257	1,704,673	890	34,494,591	44.6%	11,755,926	15.2%
mRNA-seq	48	6	170,199,865	6,635,884	1,559,173	1,853	61,466,652	36.1%	23,366,489	13.7%
RPF-seq	U	5	140,407,788	5,491,940	21,225,772	200	89,897,163	64.0%	3,879,844	2.8%
RPF-seq	0	2	54,423,427	1,458,940	6,774,540	83	37,411,067	68.7%	2,168,487	4.0%
RPF-seq	1	2	54,047,016	1,363,629	6,549,870	66	33,983,384	62.9%	1,953,449	3.6%
RPF-seq	2	2	54,548,807	1,351,618	6,192,193	56	33,048,976	60.6%	1,431,657	2.6%
RPF-seq	4	2	59,664,262	1,529,365	8,399,485	42	34,928,351	58.5%	1,747,511	2.9%
RPF-seq	12	2	46,750,727	1,322,140	1,190,058	14	39,779,904	85.1%	252,178	0.5%
RPF-seq	16	2	56,249,889	1,504,497	1,577,248	37	48,251,903	85.8%	729,631	1.3%
RPF-seq	24	2	47,938,414	1,296,573	1,309,253	19	42,101,925	87.8%	369,669	0.8%
RPF-seq	36	2	46,349,876	1,155,726	1,200,399	10	41,214,799	88.9%	493,105	1.1%
RPF-seq	48	3	77,010,524	3,753,426	14,421,763	139	51,102,057	66.4%	1,560,260	2.0%
QTI-seq	U	5	134,751,383	5,693,408	19,826,733	236	94,608,514	70.2%	3,043,977	2.3%
QTI-seq	0	2	55,679,206	1,671,563	6,101,827	110	42,197,987	75.8%	1,422,071	2.6%
QTI-seq	1	2	55,812,191	1,543,360	6,037,629	161	41,968,755	75.2%	1,320,501	2.4%
QTI-seq	2	2	61,231,630	1,733,330	6,280,581	99	45,436,578	74.2%	1,590,513	2.6%
QTI-seq	4	2	60,115,191	1,714,696	7,942,437	70	43,629,861	72.6%	1,373,425	2.3%
QTI-seq	12	2	49,678,760	1,436,592	1,419,982	14	43,562,377	87.7%	222,518	0.4%
QTI-seq	16	2	50,122,090	1,305,145	1,843,618	7	44,953,854	89.7%	251,565	0.5%
QTI-seq	24	2	44,003,422	1,085,428	1,253,684	20	40,207,605	91.4%	272,589	0.6%
QTI-seq	36	2	43,037,324	1,084,310	1,451,084	7	38,838,888	90.2%	241,690	0.6%
QTI-seq	48	3	69,420,456	3,976,950	12,005,006	141	48,050,772	69.2%	1,764,230	2.5%
sRNA-seq	U	2	45,026,294	1,936,117	13,754,089	261	18,763,999	41.7%	800,423	1.8%
sRNA-seq	0	1	22,311,040	797,096	8,853,549	175	10,844,041	48.6%	511,359	2.3%
sRNA-seq	1	1	21,665,030	790,584	8,999,321	120	10,379,598	47.9%	548,268	2.5%
sRNA-seq	2	2	53,798,253	1,575,552	16,117,463	701	30,957,957	57.5%	2,097,808	3.9%
sRNA-seq	4	2	55,796,404	1,726,186	17,476,148	540	30,557,112	54.8%	2,965,013	5.3%
sRNA-seq	12	1	17,819,476	565,180	1,492,912	55	12,049,831	67.6%	319,114	1.8%
sRNA-seq	16	1	25,740,487	629,746	2,085,204	89	15,845,985	61.6%	279,155	1.1%
sRNA-seq	24	1	10,475,478	260,339	605,435	51	6,926,281	66.1%	162,817	1.6%
sRNA-seq	36	1	24,883,176	674,072	2,950,028	87	14,128,613	56.8%	276,598	1.1%
sRNA-seq	48	1	26,820,359	1,337,294	7,645,836	58	5,869,397	21.9%	259,232	1.0%

Table R1. The number of reads discarded at pre-processing step.

The number of replicates, sequenced reads, and discarded reads at pre-processing steps are described for sequencing datasets. The column of infection status includes following information: U for uninfected, 0 for 0 hours post-infection (hpi), 1 for 1hpi, 2 for 2hpi, 4 for 4hpi, 12 for 12hpi, 16 for 16hpi, 24 for 24hpi, 36 for 36hpi, and 48 for 48hpi. The number of reads mapped to non-coding transcripts including tRNAs, rRNAs, and other non-coding RNAs is shown in the column 'step4' (blue). The number of reads discarded due to mycoplasma contamination is listed in the column 'step5' (orange). The relative fraction was calculated by dividing the number of reads by the total sequenced reads.

Cured Calu-3 samples:

assay	infection	replicates	sequenced	step1 (qualtrim)	step2 (adaptrim)	step3 (artifact)	step4 (ncRNA)		step5 (myco)	
RPF-seq	48	2	25,973,973	2,876,155	1,193,554	6	18,112,172	69.7%	938	0.0036%
RPF-seq	72	2	23,205,062	2,575,865	1,393,485	9	14,343,328	61.8%	21,312	0.0918%
QTI-seq	48	2	23,394,787	2,547,160	1,599,246	6	13,964,075	59.7%	783	0.0033%
QTI-seq	72	2	22,736,710	2,536,751	1,689,535	10	13,174,937	57.9%	16,386	0.0721%

Table R2. The number of reads discarded at preprocessing step (Cured Calu-3 cells).

The number of replicates, sequenced reads and discarded reads at pre-processing steps are described for all sequenced libraries of Calu-3 samples after treatment for mycoplasma contamination. Otherwise as in **Table R1**.

Fig. R2-5. Comparison of translome datasets before and after mycoplasma contamination.

a, Comparison of the RPF-seq (left) and QTI-seq (right) expression levels of SARS-CoV-2 ORFs before and after treatment in Calu-3 cell lines at 48 hours post-infection (hpi). The relative fraction of the expression level of each ORF was presented as a percentage.

b, Comparison of the RPF-seq (top) and QTI-seq (bottom) expression levels of human protein-coding genes before and after treatment in Calu-3 cell lines at 48 hours post-infection (hpi). Correlation coefficient (Spearman's ρ) was calculated by comparing host gene expression levels between translome samples before and after treatment for mycoplasma contamination. Both x- and y-axes represent $\log_{10}(\text{RPKM}+1)$.

c, Enrichment of RPF-seq (left) and QTI-seq (right) reads on TIS-L for Calu-3 cell line before treatment for mycoplasma infection (top) and after treatment (bottom). The 13th position (12-nt offset from the 5' end) of the reads, indicating the ribosome P-site position, was counted and calculated as the number of reads per million mapped reads (RPM). Open reading frames are depicted as three different colored bars with the dark blue bars indicating in-frame with TIS-L and the others out-of-frames.

d, Comparison of the relative fractions of RPF-seq (left) and QTI-seq (right) TIS-L reads uniquely mapped to each of gRNA and sgRNAs for Calu-3 cell line before treatment for mycoplasma infection and after treatment.

(R2-6) A main conclusion of the study is the translational utilisation of a CUG codon in the viral leader region at the 5' end of the genome. The authors note briefly that this has been seen in another study (ref 44), but it has also been documented for the coronavirus IBV (Dinan *et al.*, 2019; PMID 31243124) and in SARS-CoV-2 itself (Finkel *et al.*, 2021). I am very surprised that the latter manuscript wasn't mentioned at all in this submission!

We agree with the reviewer that the studies mentioned above should also be discussed in our paper as they both report the translation in the 5' leader region in coronaviruses and have added these two studies to the references as suggested.

Reviewer #3

The manuscript "A high-resolution temporal atlas of the SARS-CoV-2 translate and transcriptome" by Kim and colleagues provides a thorough and comprehensive analysis of viral and host transcription, and translation in Calu-3 cells during the life cycle of SARS-CoV-2 replication. The amount of data is overwhelming and of very good quality. The authors analysed several early (0, 1, 2, 3, 4 hours post infection) and late (12, 16, 24, 36 hours post infection) for mRNA-seq, RPF-seq, QTI-seq and sRNA-seq. The obtained data provide a very detailed overview on virus replication and host responses. The authors chose to depict individual observations, which make it difficult for the reader to see the overall picture. The data however are compelling and the authors may like to address the following comments.

(R3-1) As mentioned above, the manuscript would gain more clarity if some more general findings and observations are displayed. Although there're already ribosomal profiling data published, the current data set seems to be of higher quality and the authors should consider to display an overview on all SARS-CoV-2 ORFs that are translated. Although this may look very similar to published work, it would, first, illustrate the quality of the data and, second, allow the reader to get into the topic by providing a global overview on the translome.

We would like to thank the reviewer for the valuable advice. Following the reviewer's suggestion, we revised our study by including figures that prove the quality of our dataset and by displaying a global overview as follows.

First, we examined the length distribution of our RPF-seq and QTI-seq data. Regardless of the time point, sequencing method, or mapped genome, we observed a discrete peak at 28~29 nts (**Fig. R3-1a**), that is the expected size of ribosome footprints. We have also confirmed that the distribution of the reads mapped to the SARS-CoV-2 genome was similar to that of the host genome, supporting that our RPF-seq and QTI-seq reads mapped to the SARS-CoV-2 genome are not contaminated by untranslated viral RNA fragments with variable lengths.

Second, we investigated whether the triplet periodicity of the mapped position of reads is observed for RPF-seq and QTI-seq data, which is one of the major characteristics of ribosome profiling. For all the experiments that we have performed, >50% of the reads were mapped to the first position of triplet codon (**Fig. R3-1b**). This clear periodicity also confirms the enrichment of footprints of ribosomes, neither damaged by treatment of RNase I with high concentration nor contaminated by untranslated viral RNA fragments.

Third, we quantitatively assessed the relative distribution of the reads within human and viral ORFs for our datasets (**Fig. R3-1c**), which provides the global overview of the viral and host translome. While the mRNA-seq reads were evenly distributed, RPF-seq and QTI-seq reads clearly exhibited the triplet periodicity aforementioned and were enriched near the translation initiation sites of ORFs. The QTI-seq reads were more enriched in the annotated translation initiation sites for ORFs compared to RPF-seq reads, confirming the overall high quality of our QTI-seq data which captured the footprint of ribosomes initiating translation.

Forth, we have also examined our translome data and evaluated its quality based on programmed frameshifting (PRF), which is an important characteristic of coronavirus and therefore should be detected in RPF-seq dataset. While translating genomic RNA, PRF occurs to express ORF1b and is known to show 15%-75% efficiency depending on the type of virus (Plant and Dinman, 2008; Irigoyen *et al.*, 2016). When we measured the PRF efficiency by comparing the read densities between ORF1a and ORF1b, 64% efficiency was obtained on average (**Fig. R3-1d**), which is comparable to $57 \pm 12\%$ reported in a previous

study (Finkel *et al.*, 2021). The observation of general feature for coronavirus translation further supports that our RPF-seq dataset captured the true footprint of ribosomes. Thanks to the reviewer, we have been able to improve our manuscript to illustrate the high quality of our datasets and to provide a global overview on the translome by including these results.

Fig. R3-1. Overview of the SARS-CoV-2 translome dataset.

a, Distribution of read lengths for translome dataset. For RPF-seq (top) and QTI-seq (bottom) reads

mapped to the human (green) or SARS-CoV-2 genome (red), distributions of the read lengths are shown for each time point and each cell line.

b, Triplet nucleotide periodicity of translome dataset. The 13th nucleotide position of the reads mapped to the human (green) or SARS-CoV-2 (red) was counted for RPF-seq (top) and QTI-seq (bottom). The relative fractions of the reads mapped to each of three codon nucleotides are shown for each time point and each cell line. Open reading frames are depicted as three different colored bars with the darkest bars indicating in-frame and the others out-of-frames.

c, Distribution of mRNA-seq (left), RPF-seq (middle), and QTI-seq (right) reads with respect to the relative position near the start of ORF. For 4 hours post-infection (hpi) and 36 hpi, the 13th nucleotide position (12-nt offset from the 5' end) of the reads mapped to the human (green) or SARS-CoV-2 genome (red) was counted for each sequencing data. The relative fraction to the amount of reads mapped to entire ORF was calculated for each position, and the y-axis represents the average of the relative fractions for ORFs with >50 reads mapped. Open reading frames are depicted as three different colored bars with the darkest bars indicating in-frame and the others out-of-frames.

d, Efficiency of programmed ribosomal frameshifting (PRF) between ORF1a and ORF1b measured for each time point and each cell line.

(R3-2) After providing a more general overview, the reader would be prepared to "digest" the more specific findings, such as the leader TIS. This part on the TIS-L is taking a prominent part of the entire manuscript, however, the overall biological significance is not clear (see below). Therefore, an introduction with a more general overview would raise the acceptance of the reader to get into the more speculative TIS-L part.

Following the reviewer's comments **R3-1** and **R3-2**, we have improved our manuscript by providing the general findings from our dataset and global overview on the SARS-CoV-2 translome. In addition, we revised our Results section by adding the general description on SARS-CoV-2 translation and emphasizing the necessity of revealing the translational regulatory mechanism to more comprehensively understand the molecular mechanism of SARS-CoV-2 pathophysiology. We hope this would help the readers to clearly understand the biological significance of our findings including TIS-L.

(R3-3) While the TIS-L may have some role to play in SARS-CoV-2 replication, it is not clear of this mechanism is relevant (for the scope of this manuscript it is acceptable to describe the observation) for SARS-CoV-2 or other CoVs. Some statements should be adjusted since there's no evidence about the impact of the TIS-L. For example, the statement that the "TIS-L has a substantial regulatory impact on most SARS-CoV-2 ORFs..." is not supported by the provided data. It should be phased in a more speculative way. It is not clear if a 2.6- or 2-fold promotion or reduction of a reporter protein expression is meaningful for viral gene expression.

We agree that the observed change in protein expression by TIS-L (**Fig. 4f**) alone cannot fully support the actual pathophysiological impact of TIS-L. However, our results indicate that the level of translation initiation of TIS-L is even higher than that of annotated AUG for several ORFs (**Fig. 3f**), which suggests that TIS-L significantly influences the translation of nearly all viral ORFs. Therefore, we postulate that accumulation of >2-fold regulatory impact of TIS-L on all SARS-CoV-2 ORFs may play critical roles for SARS-CoV-2 pathophysiology. In response to the reviewer, we have clarified the statements mentioned

above to better support our claim regarding the potential regulatory impact of TIS-L.

(R3-4) An overview on the position of TIS-L in relation to start codons of viral ORFs on viral genomic and sg mRNAs should be provided as an overview figure panel.

Following the reviewer's suggestion, we have updated **Figure 4** by adding the illustration on the locations of TIS-L and start codon of primary ORF for each of viral gRNA and sgRNAs (**Fig. R3-4**).

Fig. R3-4. Overview on the positions of TIS-L, TRS, start codons, and annotated ORFs of SARS-CoV-2 gRNA and sgRNAs.

For each of gRNA and sgRNAs, positions of TIS-L (orange), TRS (yellow), and start codon of ORFs (green) were displayed. The reading frame of each TIS-L-initiating hypothetical ORF was shown as a green (in frame) or red (out-of-frame) dashed line compared to the reading frame of the annotated ORF.

(R3-5) Since expression of ORF10 is still under debate in the field, the authors may summarise some data regarding the detection of ORF10-containing mRNA (does it exist or not). Apparently, ORF10 is translated but there's still the question if a separate sg mRNA exists. The authors may contribute to this discussion with their data and some words in the discussion.

Following the reviewer's suggestion, we investigated whether ORF 10 sgRNA is produced by non-canonical junctions in order to help evaluate whether ORF 10 is expressed and functional. When measuring the number of junctions that occurred between TRS-L and the 5' end of ORF 10, a very small amount of the reads (<8 RPM), which corresponds to <0.02% of the total junction-spanning reads, were obtained (**Fig. R3-5, top**). Moreover, the majority of these ORF 10 non-canonical junctions occurred downstream of ORF 10, and these hypothetical ORFs were out-of-frame with ORF 10 (**Fig. R3-5, bottom**). From these observations, we conclude that the TRS-dependent junctions specific for ORF 10 sgRNA are not produced during viral discontinuous transcription. However, translation initiation at the annotated TIS and an increased level of translation at 24 and 36 hpi was clearly detected for ORF 10 both from RPF-seq and QTI-seq results (**Supplementary Fig. 8b,c**), indicating that ORF 10 might produce a functional protein. These results collectively suggest the translation of ORF 10 mainly occurs from sgRNAs of the other ORFs, rather than from its own sgRNAs, perhaps by ribosomal leaky scanning or reinitiation after the termination of translation for upstream ORFs. Thanks to the reviewer, we have improved the discussion on ORF 10 sgRNA production and translation in our manuscript by adding the result of mRNA-seq analysis. We hope it would help to resolve the controversy over ORF 10 expression and function.

Fig. R3-5. Limited production of non-canonical ORF 10 sgRNA and translation of ORF 10.

From mRNA-seq at 36 hours post-infection (hpi), the reads that include a junction located in the region between 100nt upstream and 50nt downstream of ORF 10 start position were counted and calculated as the number of reads per million mapped reads (RPM). The most abundant five junction pairs are shown with their genomic positions. ORF 10 sgRNAs created from each junction and their reading frames to ORF 10 are depicted below. Open reading frames are depicted as three different colored bars with the darkest bars indicating in-frame and the others out-of-frames with respect to ORF 10.

(R3-6) The individual figures are very data rich and some details are only rudimentary (or not) mentioned in the text. It would be good to adjust the main text to accommodate all these interesting findings.

We would like to thank the reviewer for the valuable comments. We have largely revised our manuscript to incorporate all the findings from our transcriptome and translome datasets. More detailed interpretations on the results were added into the manuscript and more detailed descriptions on the figures were added to relevant figure legends. We hope this modification improves the overall readability of our manuscript.

(R3-7) A more elaborate discussion would help the reader to judge on the impact of individual findings.

Following the reviewer's suggestion, we have largely improved the discussions on each of our findings, including the potential regulatory impact of TIS-L, the possible pathophysiological utilization of the temporal atlas of SARS-CoV-2 translome, and the overall quality assessment of our datasets.

Reviewer comments, second round –

Reviewer #1 (Remarks to the Author):

The authors have adequately addressed all of my questions.

Reviewer #2 (Remarks to the Author):

The resubmitted manuscript from Kim and colleagues contains additional information, new experiments and has broader and more thoughtful discussion. I think the manuscript is considerably improved, but there are a few points that still need addressing.

1. In my initial comments, I queried whether the quality of the ribosome profiling data was sufficiently high because the density of reads across viral ORFs 1a and 1b looked very similar, suggestive of either a programmed ribosome frameshifting efficiency approaching 100%, or inadvertent contamination of the reads with viral RNA or ribonucleoproteins. The latter artefacts are typically noticeable in low-occupancy regions of the genome (like ORF1b). The authors have indicated that this is simply due to the log scale nature of the plot, making small differences hard to spot, and this may be a reasonable explanation. Indeed, to their credit, the authors now show size length distribution and phasing plots that support the idea that the reads are genuine ribosome-protected fragments (RPFs) (although a little over trimmed) and there is some phasing of reads. However, I am surprised that the authors do not show a linear scale panel of the 1a/1b region to back up these claims. Whilst many time points show very high PRF efficiencies (>60%), which might make even linear scale comparisons difficult, the proposed PRF efficiency at the 4h and 48h post-infection time points is relatively low (20-30%). Linear plots of these time points should be provided to satisfy the reader that the reads are genuine and their levels conform to the proposed PRF efficiency.

I may appear to be over-critical about this issue, but the image of the switch of phasing of reads when ribosomes move from ORF1a to ORF1b is not particularly supportive of the authors claim that there is no problem with read quality (Fig. R2-3 panel A). If there is phasing in the datasets, then there should be a clear peak in the ORF1b phase, not three relatively even peaks in each phase within ORF1b. This is not very convincing and I hope that the authors will show the linear plots of reads across 1a and 1b to support their claims.

2. One of the important claims of the manuscript is that the Leader TIS (CUG) is functionally present on sgmRNAs, in contrast to the conclusions of Finkel et al. in a recent, related study (PMID: 32906143) who argued that most reads containing this region were almost exclusively from the genomic RNA. This has significant implications for how virus gene expression might be regulated. Two things stand out from this:

a). Within the sgmRNAs, the S ORF start codon would be in frame with the upstream leader-derived CUG TIS, thus one would expect the virus to express a significant proportion of S protein with a short N-terminal extension. Is there any evidence for such a species from mass spectroscopy studies?

b). Supporting evidence for the presence of the Leader TIS on sgmRNAs comes from identifying chimeric reads of sufficient length to span the Leader-Body junctions and be unambiguously assigned to a particular sgmRNA. Unfortunately, as the RPF read lengths were mostly <29 nt, only a few reads were long enough (a few thousand of the millions of reads in this region) for this purpose. My concern is that of the reads displayed (Fig 3c, Fig 3e, SuppFig 3c), hardly any have the CUG in the P-site of the ribosome (12nt offset to the 5' end of the read) and the few that do are those that map to the genomic RNA. This could suggest that the ribosomes from which these reads were derived are not sitting on the CUG codon. Alternatively, they must be heavily trimmed at the 5' end. It is surprising that the authors did not display those chimeric reads that map to the S sgmRNA, as this is the gene for which initiation from the CUG codon would be in frame with the

S AUG start codon.

3. Table R1 should be included in the Supplementary information and should include an extra column showing final read numbers used in the bioinformatics analysis.

4. The authors should comment somewhere in the text that the cells had mycoplasma contamination but that control experiments revealed that this did not obviously affect the datasets.

Reviewer #3 (Remarks to the Author):

The authors have invested considerable effort to address the reviewers comments and the manuscript has improved. This is a data-rich resource for future studies and will stimulate the field. Nevertheless, it should be made clear in the text that one limitation is that the work is largely descriptive and that proposed regulatory functions e. of TIS-L is still hypothetical and need further experimental confirmation in future studies. Despite this limitation, this study is a significant contribution to the field.

One minor point to address is the role of a putative sg mRNA7b. Similar as the role of ORF10 is still elusive (the authors addressed this point very well), the role of a postulated mRNA7b is not clear. There's evidence in several studies that a sg mRNA7b may exist, but read counts that are specific for the leader-body junction of a mRNA7b are very low in this manuscript and in previously published studies. Furthermore, studies that show northern blot analyses of SARS-CoV-2 RNAs, find only one band corresponding to mRNA7a. So if there's a considerable amount of mRNA7b in SARS-CoV-2 it appears to be not visible in northern blots and by RNAseq analyses read counts are always low. This raises the question if mRNA7b (if it exists) has any relevance. The authors may briefly comment in the text on the unclear role and existence of a mRNA7b.

Response to reviewers' comments

We would like to thank the reviewers for their careful and thorough evaluation of this manuscript and for the insightful comments and constructive suggestions, which helped us to improve the quality of the manuscript. In addition, we have performed more detailed analyses on the translation of ORF10 and revised our manuscript accordingly. Please see below, in blue, our detailed response to reviewers' comments.

Reviewer #1

The authors have adequately addressed all of my questions.

Reviewer #2

The resubmitted manuscript from Kim and colleagues contains additional information, new experiments and has broader and more thoughtful discussion. I think the manuscript is considerably improved, but there are a few points that still need addressing.

(R2-1) In my initial comments, I queried whether the quality of the ribosome profiling data was sufficiently high because the density of reads across viral ORFs 1a and 1b looked very similar, suggestive of either a programmed ribosome frameshifting efficiency approaching 100%, or inadvertent contamination of the reads with viral RNA or ribonucleoproteins. The latter artefacts are typically noticeable in low-occupancy regions of the genome (like ORF1b). The authors have indicated that this is simply due to the log scale nature of the plot, making small differences hard to spot, and this may be a reasonable explanation. Indeed, to their credit, the authors now show size length distribution and phasing plots that support the idea that the reads are genuine ribosome-protected fragments (RPFs) (although a little over trimmed) and there is some phasing of reads. However, I am surprised that the authors do not show a linear scale panel of the 1a/1b region to back up these claims. Whilst many time points show very high PRF efficiencies (>60%), which might make even linear scale comparisons difficult, the proposed PRF efficiency at the 4h and 48h post-infection time points is relatively low (20-30%). Linear plots of these time points should be provided to satisfy the reader that the reads are genuine and their levels conform to the proposed PRF efficiency.

I may appear to be over-critical about this issue, but the image of the switch of phasing of reads when ribosomes move from ORF1a to ORF1b is not particularly supportive of the authors claim that there is no problem with read quality (Fig. R2-3 panel A). If there is phasing in the datasets, then there should be a clear peak in the ORF1b phase, not three relatively even peaks in each phase within ORF1b. This is not very convincing and I hope that the authors will show the linear plots of reads across 1a and 1b to support their claims.

We would like to thank the reviewer for appreciating the quality of our ribosome-protected fragments (RPFs) data based on the read length distribution, phasing plots, and our response to your earlier comments. Although we observed a reduced translation of ORF 1b relative to 1a in the simplified version

of the result (**Fig. R2-3b** in the 1st revision), it was not easy to recognize the pattern of programmed ribosomal frameshifting (PRF) in read coverage plot at a single nucleotide resolution due to the sparse and uneven distribution of the reads (**Fig. R2-1**, the first row). To reliably observe PRF from RPF-seq data, relatively high density of translating ribosomes on ORFs 1a and 1b are required. Accordingly, we have decided to present coverage plots in the linear-scale of y-axis at 4 and 12 hpi, where the translation efficiency (TE) level of ORFs 1a and 1b (**Fig. 2b**) are relatively high, rather than 48 hpi, where the translation efficiency (TE) of ORFs 1a and 1b is extremely low ($\log_{10}TE < -2.0$). To help more clearly visualize the PRF, we have computed moving average of the read coverage within the region of ORFs 1a and 1b (**Fig. R2-1**, from the second row). Thanks to the reviewer, we have observed a clearer difference in RPF coverages between ORFs 1a and 1b and, included this result in our manuscript (**Supplementary Fig. 2i**).

Figure R2-1. RPF-seq coverage plots across ORFs 1a and 1b (Supplementary Fig. 2i in the revised manuscript).

Read coverages of ORFs 1a and 1b in RPF-seq samples at 4 hpi and 12 hpi. Read coverage at a nucleotide resolution (the first row) and smoothed read coverage (the second row) with moving average of a 4-kb sliding window are shown. The x-axis represents the nucleotide position of the SARS-CoV-2 genome.

(R2-2) One of the important claims of the manuscript is that the Leader TIS (CUG) is functionally present on sgmRNAs, in contrast to the conclusions of Finkel et al. in a recent, related study (PMID: 32906143) who argued that most reads containing this region were almost exclusively from the genomic RNA. This has significant implications for how virus gene expression might be regulated. Two things stand out from this:

(R2-2-1) Within the sgmRNAs, the S ORF start codon would be in frame with the upstream leader-derived CUG TIS, thus one would expect the virus to express a significant proportion of S protein with a short N-terminal extension. Is there any evidence for such a species from mass spectroscopy studies?

The S protein consists of S1 and S2 subunits and the S1 subunit has a signal peptide at its N-terminal

end (Xia, *Viruses*, 2021), which is cleaved after the ribosome containing the nascent chain is guided to the ER (Kriegler *et al.*, *Cell Chemical Biology*, 2018). The signal sequence of the spike protein is predicted to be the first 13 amino acids (MFVFLVLLPLVSS) (Xia, *Viruses*, 2021), and this result was reproduced when the signal sequence and its cleavage site was predicted using SignalP 5.0 (Armenteros *et al.*, *Nature Biotechnology*, 2019) (**Fig. R2-2-1, top**). When the signal sequence and its cleavage site were predicted with the extended N-terminal sequence assuming that the translation were to initiate from TIS-L, a similar result was obtained (**Fig. R2-2-1, bottom**). Therefore, although the TIS-L initiated protein translation may produce S protein with a short N-terminal extension (LFSKRT), it is expected to be cleaved with the signal sequence and does not appear to affect the final mature product of spike protein. Consistently, no N-terminal extended form of S protein has been reported so far for SARS-CoV-2 (Watanabe *et al.* 2020, *Science*; Bouhaddou *et al.*, 2020, *Cell*; Dollman *et al.*, 2020, *ACS*) and SARS-CoV (Ying *et al.*, 2004, *Proteomics*; Ritchie *et al.*, 2010, *Virology*; Krokhin *et al.*, 2003, *MCP*). Therefore, the translation initiation from TIS-L appears to largely increase the translation efficiency of the S ORF (**Fig. 3g**), without affecting the N-terminal sequence of S protein. Thanks to the reviewer, we have improved our manuscript by adding this result (**Supplementary Fig. 7h**).

Figure R2-2-1. Predicted signal peptide and its cleavage site of the spike protein. (Supplementary Fig. 7h in the revised manuscript)

The signal peptide and its cleavage site of the spike protein assuming translation is either initiated from AUG (top) or CUG (bottom) were predicted using SignalP-5.0. The y-axis represents the probability of the site being the signal peptide. The predicted cleavage sites of the signal peptide are depicted as black triangles.

(R2-2-2) Supporting evidence for the presence of the Leader TIS on sgmRNAs comes from identifying chimeric reads of sufficient length to span the Leader-Body junctions and be unambiguously assigned to a particular sgmRNA. Unfortunately, as the RPF read lengths were mostly <29 nt, only a few reads were long enough (a few thousand of the millions of reads in this region) for this purpose. My concern is that of the reads displayed (Fig 3c, Fig 3e, SuppFig 3c), hardly any have the CUG in the P-site of the ribosome (12nt offset to the 5' end of the read) and the few that do are those that map to the genomic RNA. This could suggest that the ribosomes from which these reads were derived are not sitting on the CUG codon.

Alternatively, they must be heavily trimmed at the 5' end. It is surprising that the authors did not display those chimeric reads that map to the S sgRNA, as this is the gene for which initiation from the CUG codon would be in frame with the S AUG start codon.

As the reviewer pointed out, most of the reads mapped to TIS-L are multiple mapped due to the short read lengths and the location of TIS-L and TRS on the genome. To overcome this limitation, we collected a subgroup of reads that are mapped to TIS-L and its next codons (59th-64th nt positions) and that span to at least 3 nts downstream of TRS. As a result, majority of the reads displayed in **Fig. 3c** were derived from the codon right next to the TIS-L. Although these reads do not exactly have TIS-L in the P-site, allowing these reads can be justified because they also follow the triplet nucleotide periodicity of TIS-L (**Fig. 3b**), and therefore are considered to include translation signals starting from TIS-L.

In order to alleviate the reviewer's concern, we repeated the analyses of **Fig. 3c, e** with a subgroup of long reads that span to at least 3 nts downstream of TRS and exactly mapped to the C nucleotide position of TIS-L. Although the number of the reads is fairly small, <3% of the reads were mapped to gRNA for both in high (**Fig. R2-2-2a**) and low (**Fig. R2-2-2b**) RNase I concentration, consistent with the estimation from our current method. Thanks to the reviewer, we have included this result in our manuscript and improved our **Methods** section by adding a more detailed description and justification regarding the estimation of the relative fraction of TIS-L reads. We have also revised **Fig. 3c, e**, and related supplementary figures by additionally displaying the TIS-L reads mapped to S sgRNA.

Fig. R2-2-2. 3' ends of RPF-seq reads mapped on TIS-L with a subset of RPF-seq reads more confidently mapped to TIS-L (Supplementary Fig. 3e, f in the revised manuscript).

For 48 hpi (**a**), a subset of long RPF-seq reads that the 13th position (12-nt offset from the 5' end) of the reads mapped to C nucleotide position of TIS-L were collected (see **Methods**). The alignments of those uniquely mapped reads to the gRNA or sgRNAs are displayed and the corresponding read counts with the relative fraction of reads mapped to each ORF are shown in parentheses. Similar analysis was performed for RPF-seq dataset with low RNase I concentration (**b**). For the low RNase I concentration dataset, 13-nt offset from the 5' end of the reads was used for the analysis, which showed the best triplet nucleotide periodicity and enrichment of reads at start codons.

(R2-3) Table R1 should be included in the Supplementary information and should include an extra column showing final read numbers used in the bioinformatics analysis.

We have added the column for the numbers of remaining reads after pre-processing steps in our revised manuscript.

assay	infection	replicates	sequenced	step1 (qualtrim)	step2 (adaptrim)	step3 (artifact)	step4 (ncRNA)	step5 (myco)	remaining
sRNA-seq	U	2	45,026,294	1,936,117	13,754,089	261	18,763,999 41.7%	800,423 1.8%	9,771,405
sRNA-seq	0	1	22,311,040	797,096	8,853,549	175	10,844,041 48.6%	511,359 2.3%	1,304,820
sRNA-seq	1	1	21,665,030	790,584	8,999,321	120	10,379,598 47.9%	548,268 2.5%	947,139
sRNA-seq	2	2	53,798,253	1,575,552	16,117,463	701	30,957,957 57.5%	2,097,808 3.9%	3,048,772
sRNA-seq	4	2	55,796,404	1,726,186	17,476,148	540	30,557,112 54.8%	2,965,013 5.3%	3,071,405
sRNA-seq	12	1	17,819,476	565,180	1,492,912	55	12,049,831 67.6%	319,114 1.8%	3,392,384
sRNA-seq	16	1	25,740,487	629,746	2,085,204	89	15,845,985 61.6%	279,155 1.1%	6,900,308
sRNA-seq	24	1	10,475,478	260,339	605,435	51	6,926,281 66.1%	162,817 1.6%	2,520,555
sRNA-seq	36	1	24,882,476	674,072	2,050,038	87	14,128,642 56.8%	276,508 1.1%	6,952,778
QTI-seq	10	2	50,122,090	1,305,145	1,045,016	7	44,555,804 89.1%	231,365 0.5%	1,707,901
QTI-seq	24	2	44,003,422	1,085,428	1,253,684	20	40,207,605 91.4%	272,589 0.6%	1,184,096
QTI-seq	36	2	43,037,324	1,084,310	1,451,084	7	38,838,888 90.2%	241,690 0.6%	1,421,345
QTI-seq	48	3	69,420,456	3,976,950	12,005,006	141	48,050,772 69.2%	1,764,230 2.5%	3,623,357

Supplementary Table 1. The number of reads at pre-processing step.

The number of replicates, sequenced reads, and discarded reads at pre-processing steps are described for sequencing datasets. The column of infection status includes following information: U for uninfected, 0 for 0 hours post-infection (hpi), 1 for 1hpi, 2 for 2hpi, 4 for 4hpi, 12 for 12hpi, 16 for 16hpi, 24 for 24hpi, 36 for 36hpi, and 48 for 48hpi. The number of reads mapped to non-coding transcripts including tRNAs, rRNAs, and other non-coding RNAs is shown in the column 'step4'. The number of reads mapped to the mycoplasma genomes and thus discarded is listed in the column 'step5'. The relative fraction was calculated by dividing the number of reads by the total sequenced reads. The number of remaining reads after 5 pre-processing steps is shown in the column 'remaining'. These reads were used for alignment and further analyses.

(R2-4) The authors should comment somewhere in the text that the cells had mycoplasma contamination but that control experiments revealed that this did not obviously affect the datasets.

Following the reviewer's suggestion, we have revised our manuscript by adding statements and figures for our consistent result regardless of mycoplasma contamination. Thanks to the reviewer, we were able to alleviate the potential readers' concern of mycoplasma contamination affecting the outcome of our study.

Reviewer #3

(R3-1) The authors have invested considerable effort to address the reviewers comments and the manuscript has improved. This is a data-rich resource for future studies and will stimulate the field. Nevertheless, it should be made clear in the text that one limitation is that the work is largely descriptive and that proposed regulatory functions e. of TIS-L is still hypothetical and need further experimental confirmation in future studies. Despite this limitation, this study is a significant contribution to the field.

Following the reviewer's suggestion, we have revised our manuscript by adding statements about the limitation of our study and that further research is needed to more clearly confirm the regulatory function of TIS-L and to reveal its mechanism of action.

(R3-2) One minor point to address is the role of a putative sg mRNA7b. Similar as the role of ORF10 is still elusive (the authors addressed this point very well), the role of a postulated mRNA7b is not clear. There's evidence in several studies that a sg mRNA7b may exist, but read counts that are specific for the leader-body junction of a mRNA7b are very low in this manuscript and in previously published studies. Furthermore, studies that show northern blot analyses of SARS-CoV-2 RNAs, find only one band corresponding to mRNA7a. So if there's considerable amount of mRNA7b in SARS-CoV-2 it appears to be not visible in northern blots and by RNAseq analyses read counts are always low. This raises the question if mRNA7b (if it exists) has any relevance. The authors may briefly comment in the text on the unclear role and existence of a mRNA7b.

As the reviewer pointed out, the role of putative ORF 7b sgRNA is unclear due to its low level of production and inability to translate ORF 7b. We agree that the functionality of ORF 7b sgRNA should be characterized in the future, and therefore we added a brief comment on this issue in our manuscript.

Third round comments

Reviewer #2 (Remarks to the Author):

The authors have satisfied my concerns and have further improved the manuscript.